# P-MapNet: Far-seeing Map Constructor Enhanced by both SDMap and HDMap Priors

## Abstract

Autonomous vehicles are gradually entering city roads today, with the help of high-definition maps (HDMaps). However, the reliance on HDMaps prevents autonomous vehicles from stepping into regions without this expensive digital infrastructure. This fact drives many researchers to study online HDMap construction algorithms, but the performance of these algorithms is still unsatisfying. We present P-MapNet, in which the letter P highlights the fact that we focus on incorporating map priors to improve model performance. Specifically, we exploit priors in both SDMap and HDMap. On one hand, we extract *weakly aligned* SDMap from OpenStreetMap, and encode it as an alternative conditioning branch. Despite the misalignment challenge, our attention-based architecture adaptively attends to relevant SDMap skeletons and significantly improves performance. On the other hand, we exploit a masked autoencoder to capture the prior distribution of HDMap, which can serve as a refinement module to mitigate occlusions and artifacts. Both priors lead to performance improvements, especially in farther regions. We benchmark on the nuScenes dataset, demonstrating $13.46\%$ mIoU margin over the baseline. Codes and models will be publicly available.

## 1 Introduction

While we still don't know the ultimate answer to fully autonomous vehicles that can run smoothly in each and every corner of the earth, the community does have seen some impressive milestones, e.g., robotaxis are under steady operation in some big cities now. Yet current autonomous driving stacks heavily depend on an expensive digital infrastructure: HDMaps. With the availability of HDMaps, local maneuvers are reduced to lane following and lane changing coupled with dynamic obstacle avoidance, significantly narrowing down the space of decision making. But the construction of HDMaps, which is shown in the left-top panel of Fig. 1, is very cumbersome and expensive. And what's worse, HDMaps cannot be constructed for good and all, because they must be updated every three months on average. It is widely recognized that reducing reliance on HDMaps is critical.

Thus, several recent methods (Li et al., 2022; Liu et al., 2022a) construct HDMaps using multi-modal online sensory inputs like LiDAR point clouds and panoramic multi-view RGB images, and a conceptual illustration of this paradigm is given in the left-middle panel of Fig. 1. Despite promising results achieved by these methods, online HDMap constructors still report limited quantitative metrics and this study focuses on promoting their performance using priors. Specifically, two sources of priors are exploited: SDMap and HDMap, as demonted in the left-botoom panel of Fig. 1.

**SDMap Prior.** Before the industry turns to build the digital infrastructure of HDMaps in a large scale, Standard Definition Maps (SDMaps) have been used for years and largely promoted the convenience of our everyday lives. Commercial SDMap applications provided by Google or Baidu help people navigate big cities with complex road networks, telling us to make turns at crossings or merge into main roads. SDMaps are not readily useful for autonomous cars because they only provide centerline skeletons (noted as SDMap Prior in the left-bottom panel of Fig. 1). So we aim to exploit SDMap priors to build better online HDMap construction algorithms, which can be intuitively interpreted as *drawing* HDmaps around the skeleton of SDMaps. However, this intuitive idea faces a primary challenge: misalignment. Per implementation, we extract SDMaps from OpenStreetMap using GPS signals but unfortunately they are, at best, weakly aligned with the ground truth HDMap in a certain scenario. An illustration is given in the right panel of Fig. 1, noted as

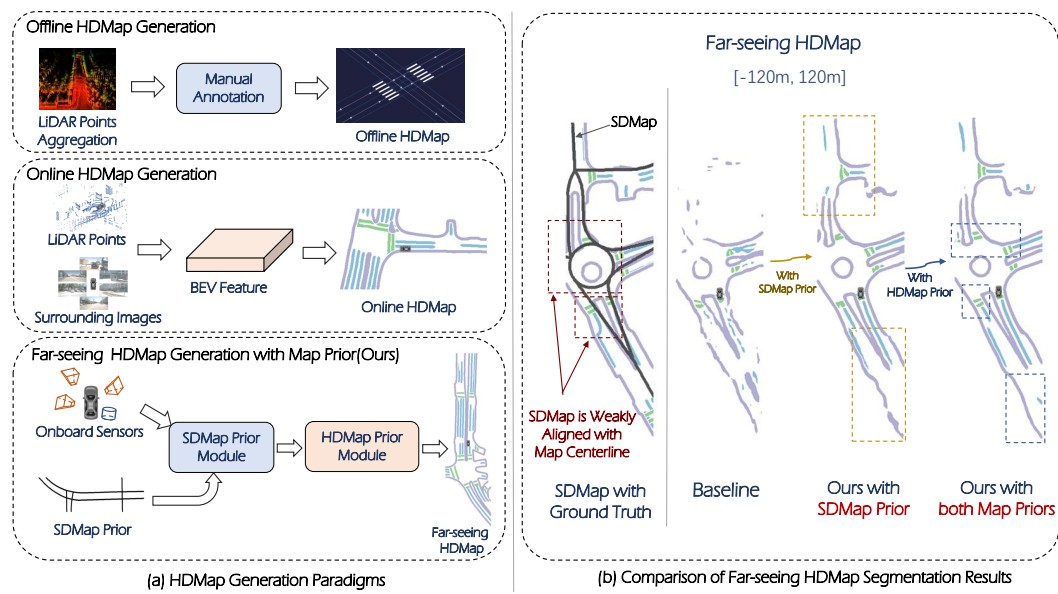

Figure 1: Left: Since offline HDMap construction is cumbersome and expensive, people are pursuing online HDMap construction algorithms and our P-MapNet improves them using both SDMap and HDMap priors. Right: Despite the misalignment between SDMaps and HDMaps, our P-MapNet can significantly improve map construction performance, especially on the far side.

SDMap with Ground Truth. To this end, we leverage an attention based neural network architecture that adaptively attends to relavent SDMap features and successfully improve the performance by large margins in various settings (see Table. 2 and Table. 12).

**HDMap Prior.** Although useful, SDMap priors cannot fully capture the distribution of HDMap output space. As noted by Ours with SDMap Prior in the right panel of Fig. 1, HDMap generation results are broken and unnecessarily curved. This is credited to the fact that our architecture is, like prior methods, designed in a BEV dense prediction manner and the structured output space of BEV HDMap cannot be guaranteed. As such, HDMap prior comes to the stage as a solution and the intuition is that if the algorithm models the structured output space of HDMaps explicitly, it can naturally correct these unnatural artifacts (i.e., broken and unnecessarily curved results mentioned above). On the implementation side, we train a masked autoencoder (MAE) on a large set of HDMaps to capture the HDMap prior and used it as a refinement module. As noted by Ours with both Map Priors in the right panel of Fig. 1, our MAE successfully corrects aforementioned issues.

**P-MapNet as a far-seeing solution.** A closer look at the positive margins brought by incorporating priors reveals that P-MapNet is a far-seeing solution. As shown in the right panel of Fig. 1, after incorporating the SDMap prior, missing map elements far from the ego vehicle (denoted by the car icon) are successfully extracted. This is understandable as the road centerline skeletons on the far side are already known in the SDMap input. Meanwhile, the HDMap prior brings improvements in two kinds of regions: crossings with highly structured repetitive patterns and lanes on the far side. This is credited to the fact that our masked autoencoder can incorporate the priors about how typical HDMaps look like, e.g., lanes should be connected and largely straight and crossings are drawn in a repetitive manner. As the later experiments in Table. 2 demonstrate, positive margins steadily grow along with the sensing range. We believe P-MapNet, as a far-seeing solution, is potentially helpful in deriving more intelligent decisions that are informed of maps on the far side.

In summary, our contributions are three-fold: (1) We incorporate SDMap priors into online map constructors by attending to weakly aligned SDMap features and achieve significant performance improvements; (2) We also incorporate HDMap priors using a masked autoencoder as a refinement module, correcting artefacts that deviate the structured output space of HDMaps; (3) We achieve state-of-the-art results on public benchmarks and present in-depth ablative analyses that reveal the mechanisms of P-MapNet. For example, P-MapNet is a far-seeing solution.

## 2 RELATED WORK

### 2.1 ONLINE HD MAP CONSTRUCTION

Traditionally, HD maps are manually annotated offline, combining point cloud maps via SLAM algorithms (Bao et al., 2022; Houston et al., 2021) for high accuracy but at a high cost and without real-time updates. In contrast, recent efforts have focused on utilizing onboard sensors for the efficient and cost-effective generation of online HD maps Philion & Fidler (2020); Saha et al. (2022); Li et al. (2022); Dong et al. (2022); Liu et al. (2022a); Liao et al. (2023). HDMapNet (Li et al., 2022) employs pixel-wise segmentation and heuristic post-processing, evaluating results with Average Precision (AP) and Intersection over Union (IoU). More recent approaches (Liao et al., 2022; Liu et al., 2022a; Ding et al., 2023; Yuan et al., 2023; Zhang et al., 2023), have adopted end-to-end vectorized HD map construction techniques, leveraging Transformer architectures (Vaswani et al., 2017). However, these methods rely solely on onboard sensors and may face limitations in handling challenging environmental conditions, such as occlusions or adverse weather.

### 2.2 LONG-RANGE MAP PERCEPTION

To enhance the practicality of HD maps for downstream tasks, some studies aim to extend their coverage to longer perception ranges. SuperFusion (Dong et al., 2022) combines LiDAR point clouds and camera images for depth-aware BEV transformation, yielding forward-view HD map predictions up to $90\ m$. NeuralMapPrior Xiong et al. (2023) maintains and updates a global neural map prior, enhancing online observations to generate higher-quality, extended-range HD map predictions. Gao et al. (2023) proposes using satellite maps to aid online map construction. Features from onboard sensors and satellite images are aggregated through a hierarchical fusion module to obtain the final BEV features. MV-Map (Xie et al., 2023) specializes in offline, long-range HD map generation. It aggregates all relevant frames during traversal and optimizes neural radiance fields for improved BEV feature construction.

## 3 SDMAP GENERATION

In this section, we outline our approach to generate weakly-aligned SDMap priors by leveraging OpenStreetMap (OSM) data and the GPS signals derived from autonomous vehicles. We specifically employ the nuScenes (Caesar et al., 2020) and Argoverse2 (Wilson et al., 2021) datasets as the primary basis for our research, as these datasets hold a prominent position within the autonomous driving domain. These datasets are richly equipped with sensors, including cameras, LiDAR, radars, GPS, and IMU, and they offer comprehensive map annotations such as roads, sidewalks, and lanes.

However, it's important to note that these datasets do not include the corresponding SDMap information for the captured regions. To address this limitation, we leverage Open-StreetMap (Haklay & Weber, 2008) to obtain the relevant SDMap data for these regions.

More specifically, our SDMap generation procedure involves two steps. The first step centers around the **alignment** of OSM[1] data with the operational regions of autonomous vehicles. Here we take the nuScenes dataset as an illustrative example, but it's vital to highlight that this procedure is compatible with all running vehicles with on-board GPS sensor. Each sub-map annotation of the nuScenes dataset uses a unique coordinate system, which takes the southwest corner of the captured region as the original point and adopts the WGS 84 Web Mercator projection, akin to the one utilized by

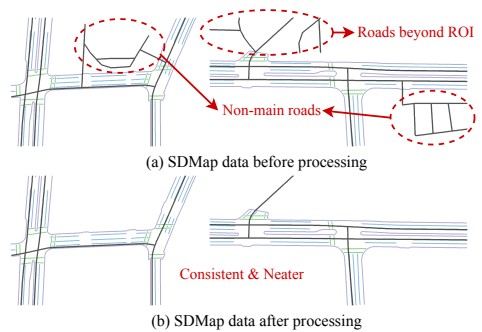

(a) SDMap data before processing

(b) SDMap data after processing

Figure 2: **The generated SDMap data** covered most of the main roads, as they are usually not easily changed. The red box in the figure shows that the secondary roads, such as living streets, or roads beyond RoI were filtered out. The final processed SDMap data has better consistency with HDMap data.

---

[1]https://www.openstreetmap.org/

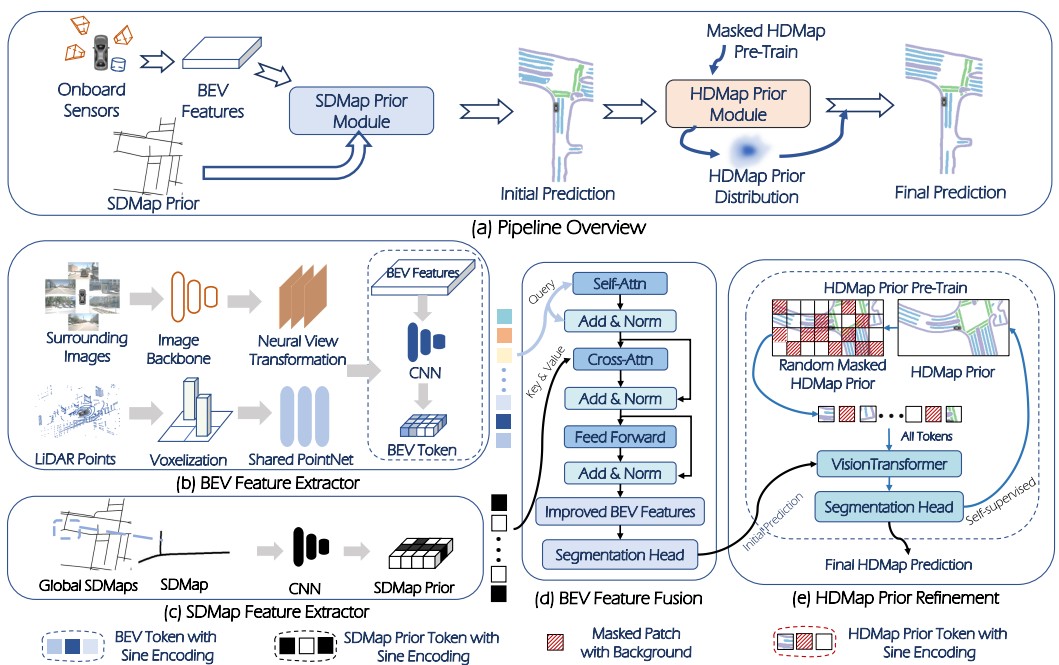

Figure 3: **P-MapNet overview.** P-MapNet is designed to accept either surrounding cameras or multi-modal inputs. It processes these inputs to extract sensors features and SDMap priors features, both represented in the Bird's Eye View (BEV) space. These features are then fused using an attention mechanism and subsequently refined by the HDMap prior module to produce results that closely align with real-world map data.

Google Maps. By downloading and subsequently projecting the OSM data onto this coordinate system, we effectively achieve alignment of OSM to the active operational region, laying the groundwork for subsequent processing.

Then, we transition to the SDMap **extraction and refinement** phase. As our primary objective is to obtain an urban road map for precise pilot planning, we begin by extracting the road map from the comprehensive OSM data, which encompasses a myriad of details including buildings and terrains. However, as illustrated in Fig. 2(a), the extracted data can sometimes feature duplicated road segments or roads that fall outside our Region of Interest (RoI). Incorporating such data in our final SDMap might introduce unwanted noise, undermining the precision we seek. To ensure the accuracy of our SDMap, we adopted a systematic filtering process. Initially, we identified trunk roads (often referred to as main roads) by filtering through the annotation category tags within the OSM data. Subsequently, with reference to the map annotation data available in the datasets, we intersected the road polygon sections from the annotation with the extracted SDMap. Only the sections with an intersection are retained, which effectively filtered out areas outside the RoI. The post-filtering results can be visualized in Fig. 2(b). To make the SDMap more usable in subsequent phases, we converted the finalized SDMap into a rasterized format, intended to act as foundational SDMap priors.

## 4 METHOD

### 4.1 ARCHITECTURE

In this section, we introduce **P-MapNet**, a novel and efficient framework for *far-seeing* perception through the seamless integration of SDMap and HDMap priors. As illustrated in Fig. 3, our approach comprises four distinct modules. The first step involves extracting bird-eye-view (BEV) features from on-board sensors. Specifically, we process surrounding images from the camera and point cloud data from the lidar through the BEV feature extractor, as detailed in Fig. 3(b). Concurrently, SDMap priors are derived using a ConvNet, as shown in Fig. 3(c). Upon obtaining the two feature maps, the immediate challenge is their integration. Merely concatenating them is suboptimal due to

the spatial misalignment of the SD priors. To address this, we employ a transformer-based module for feature alignment and fusion, as depicted in Fig.3(d) and detailed in Sec. 4.2, to produce the enhanced BEV features, which are subquently decoded by the segmentation head to produce initial prediction. Finally, the initial prediction undergoes refinement through the HDMap prior refinement module, as shown in Fig. 3(e) and elaborated in Sec. 4.3, producing final predictions that are more in tune with the real-world semantic distribution.

**BEV Feature Extractor.** We start by introducing the BEV feature extractor in Fig. 3(b). The BEV feature extractor ingests two types of on-board sensor data as input, the surrounding images and the LiDAR point cloud, which are formally denoted a $\{\mathcal{I}_i \mid i = 1, 2, \cdots N\}$ and $\mathcal{P}$, respectively. The initial step is the extraction of surrounding image features, represented as $\mathcal{F}_{I_i}$, utilizing a shared network such as ResNet (He et al., 2016) or EfficientNet (Tan & Le, 2019). Drawn inspiration from HDMapNet (Li et al., 2022), we consequently transform these image feature maps from their native perspective view into the BEV space, denoted as $\mathcal{B}_{\text{camera}} \in \mathbb{R}^{H \times W \times C_{\text{cam}}}$, where $H$ and $W$ are the height and width of the BEV grid, respectively, and $C_{\text{cam}}$ is the feature dimensionality. As for the LiDAR points $\mathcal{P}$, we use Pointpillars (Lang et al., 2019) to extract LiDAR features $\mathcal{B}_{\text{lidar}} \in \mathbb{R}^{H \times W \times C_{\text{lidar}}}$. Finally, we concatenate the BEV features of camera and lidar to get the comprehensive BEV feature maps from all sensors, which are denoted as $\mathcal{B}_{\text{sensors}} \in \mathbb{R}^{H \times W \times C}$.

**Segmentation Head and Losses.** For the extraction of semantic details from the enhanced BEV feature map, we utilize a decoder head. We opt for a fully convolutional network with dense prediction, similar to Li et al. (2022), for the segmentation of map elements, enabling the use of conventional 2D positional encoding during refinement. Our segmentation head contains a total of three branches, semantic segmentation, instance embedding prediction and direction prediction. The semantic segmentation branch classifies various map element types, including *dividers*, *sidewalks*, and *boundaries*. The instance embedding branch aids in predicting lanes at an instance level. To facilitate the creation of a vectorized HDMap during post-processing, the direction prediction branch determines the orientation of each lane pixel, ensuring a coherent linkage between pixels.

To train these three distinct branches, we introduce three specialized kinds of loss functions. For the semantic branch, we employ the cross-entropy loss, denoted as $\mathcal{L}_{\text{seg}}$. Drawing inspiration from De Brabandere et al. (2017), the instance embedding prediction is supervised by a clustering loss, represented as $\mathcal{L}_{\text{ins}}$. For direction prediction, we categorize the results into 36 classes, each corresponding to a 10-degree interval, thereby allowing the use of a cross-entropy loss, symbolized by $\mathcal{L}_{\text{dir}}$. The final loss, $\mathcal{L}$, is a weighted aggregation of the aforementioned losses, expressed as:

$$\mathcal{L} = \lambda_{\text{seg}} \mathcal{L}_{\text{seg}} + \lambda_{\text{ins}} \mathcal{L}_{\text{ins}} + \lambda_{\text{dir}} \mathcal{L}_{\text{dir}} \tag{1}$$

The coefficients $\lambda_{\text{seg}}$, $\lambda_{\text{ins}}$, and $\lambda_{\text{dir}}$ serve to balance the different loss components.

## 4.2 SDMAP PRIOR MODULE

Given the intrinsic challenges associated with onboard sensor perception, such as distant road invisibility, occlusion, and adverse weather conditions, the need to incorporate SDMap prior becomes crucial, as SDMap provides a stable and consistent representation of the environment. However, after extraction and processing, the rasterized SDMap priors may face a spatial misalignment, where the SDMap prior doesn't align precisely with the current operational location, often resulting from inaccurate GPS signals or rapid vehicle movement. Such misalignment renders the straightforward method of directly concatenating BEV features with SDMap features in the feature dimension ineffective. To tackle this challenge, we adopt a multi-head cross-attention module. This allows the network to utilize cross attention to determine the most suitably aligned location, thereby effectively enhancing the BEV feature with the SDMap prior.

Specifically, as illustrated in Fig. 3(b), we initially utilize a convolutional network to downsample the BEV features, denoted as $\mathcal{B}_{\text{sensors}}$. This not only averts excessive memory consumption on low-level feature maps but also partially alleviates the misalignment between the image BEV features and the LiDAR BEV features. The downsampled BEV features (BEV Token in Fig. 3(b)) are represented as $\mathcal{B}_{\text{small}} \in \mathbb{R}^{\frac{H}{d} \times \frac{W}{d} \times C}$, where $d$ is the downsampling factor. These features, combined with sine positional embedding, are processed through the multi-head self-attention, resulting in the initial BEV queries $\mathcal{Q}_{\text{bev}}$.

The associated SDMap, $\mathcal{M}_{\text{sd}}$, undergoes processing via a convolutional network in conjunction with sine positional embedding, producing the SDMap prior tokens $\mathcal{F}_{\text{sd}}$, as shown in Fig. 3(c). Subsequently, the multi-head cross-attention is deployed to enhance the BEV features by integrating the information from SDMap priors. The formal representation is,

$$\mathcal{Q}' = \text{Concat}\left(\text{CA}_1(\mathcal{Q}_{\text{bev}}, \mathcal{F}_{\text{sd}}, \mathcal{W}_1), \ldots, \text{CA}_m(\mathcal{Q}_{\text{bev}}, \mathcal{F}_{\text{sd}}, \mathcal{W}_m)\right),$$
$$\mathcal{B}_{\text{improved}} = \text{layernorm}\left(\mathcal{Q}_{\text{bev}} + \text{Dropout}(\text{Proj}(\mathcal{Q}'))\right), \tag{2}$$

where the $\text{CA}_i$ is the $i$-th single head cross-attention, $m$ is the number of head, $\mathcal{W}$ is the parameter set to compute the query, key and value embeddings, $\text{Proj}$ is a projection layer and $\mathcal{B}_{\text{improved}}$ represents the resized BEV feature derived from the multi-head cross-attention that incorporates the SDMap prior. Subsequentially, the improved bev features pass through a segmentation head to get the initial HDMap element prediction, denoted as $X_{\text{init}} \in \mathbb{R}^{H \times W \times (N_c+1)}$. Here, the $(N_c + 1)$ channels denote the total number of map element classes, including an additional background class.

## 4.3 HDMAP PRIOR REFINEMENT

While real roads typically exhibit continuous and consistent characteristics, current road segmentation models (Li et al., 2022; Dong et al., 2022; Xiong et al., 2023; Zhu et al., 2023) predict on a pixel-by-pixel basis. As illustrated in Fig. 1, the initial predictions from the segmentation head display gaps such as absent sidewalks and broken lane lines. Such inconsistencies make the outcomes unsuitable for autonomous driving applications. To better incorporate the HDMap prior of road continuity, we introduce a ViT (Dosovitskiy et al., 2020)-style HDMap prior refinement module, as shown in Fig. 3(e), to refine the initial prediction via modeling global correlations. Further, we lean on a masked-autoencoder (MAE) pretraining methodology, inspired by (He et al., 2022), for the pretraining of our refinement network.

Specifically, our refinement module, denoted as $\mathcal{H}$, takes $X_{\text{init}}$ as input, and outputs the refined results: $X_{\text{refined}} = \mathcal{H}(X_{\text{init}})$, where $X_{\text{refined}}$ maintains the same dimensions as $X_{\text{init}}$. Similar to the standard ViT, during the refinement phase, the $X_{\text{init}}$ is initially divided into non-overlapping patches and embedded by a linear projection with positional embeddings, followed by a sequence of Transformer blocks. To obtained the refined results, we first upsample the embedded features back to size of $\mathbb{R}^{H \times W \times D}$, where $D$ is feature dimension, and then use the segmentation head for prediction.

In the pretraining phase, we begin by cropping the HDMap data to match the sensor perception range, resulting in a sample denoted as $X_{\text{map}} \in \mathbb{R}^{H \times W \times (N_c+1)}$. We then divide the $X_{\text{map}}$ into a sample of size $h \times w$:

$$X_{\text{map}} = \left\{ x_i \in \mathbb{R}^{h \times w \times (N_c+1)} \mid i = 1, 2, \ldots, \lfloor H/h \rfloor \cdot \lfloor W/w \rfloor \right\} \tag{3}$$

where $\lfloor \cdot \rfloor$ denotes rounding down. A random portion of $X_{\text{map}}$ is selected to be masked, noted as $\{x_i^m\}$, and the remainder are visible patches $\{x_i^v\}$. We convert all the categories within $\{x_i^m\}$ to background categories to achieve effective masking, and denote the result as $\{\bar{x}_i^m\}$. Finally, we obtain the input for the pretraining phase by merging $\{\bar{x}_i^m\}$ and $\{x_i^v\}$, formulated as $X_{\text{masked}} = \{\bar{x}_i^m\} \bigcup \{x_i^v\}$, with the ground truth $X_{\text{map}} = \{x_i^m\} \bigcup \{x_i^v\}$. This random masking approach bolsters the network's ability to capture HDMap priors more efficiently. To guarantee that the reconstructed output aligns well with the original semantic annotation $X_{\text{map}}$, training is facilitated using the cross-entropy loss function.

# 5 EXPERIMENTS

## 5.1 DATASET AND METRICS

We evaluate P-MapNet on two popular datasets in autonomous driving research, nuScenes (Caesar et al., 2020) and Argoverse2 (Wilson et al., 2021). The nuScense contains 1000 scenes with six surrounding cameras and a LiDAR. And the Argoverse2 contains 1000 3D annotated scenarios in six cities. We focus on the tasks of semantic HDMap segmentation and instance detection. Similar to HDMapNet (Li et al., 2022), we conduct an evaluation of three map elements: lane dividers, pedestrian crossings, and road boundaries.

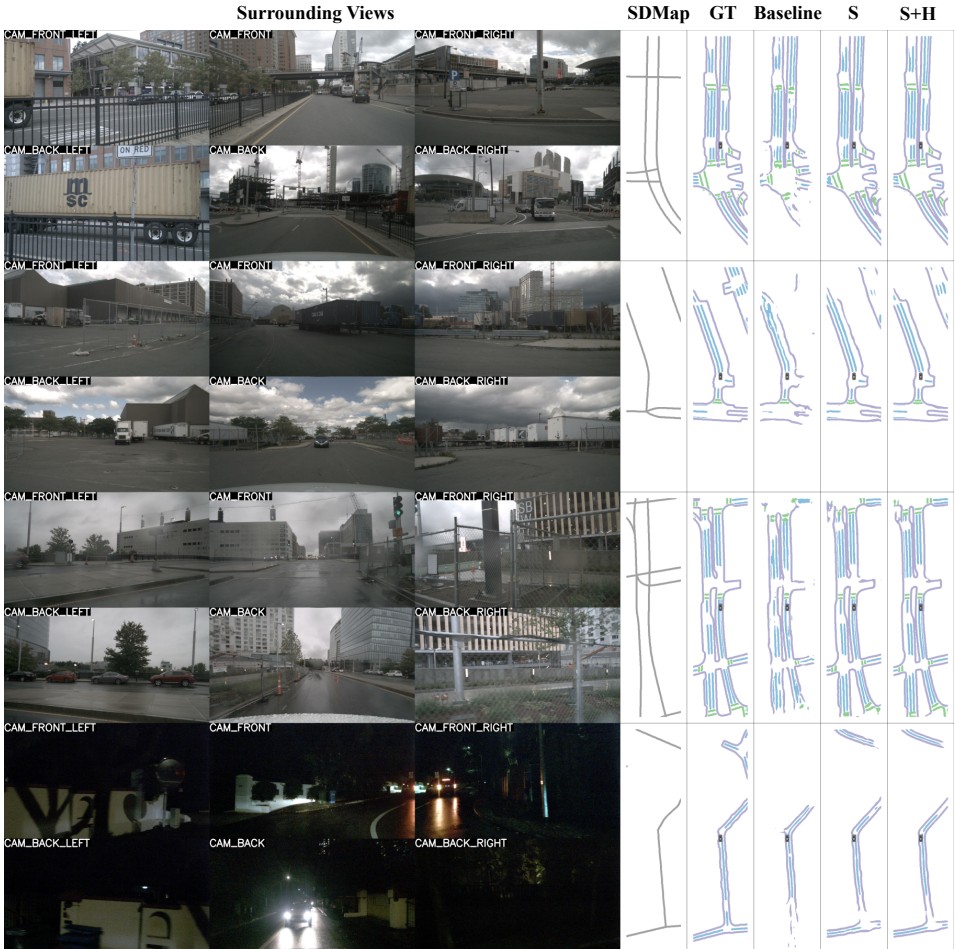

Figure 4: **Qualitative results on the nuScenes val set.** We conducted a comparative analysis within a range of 240m×60m, utilizing C+L as input. In our notation, "S" indicates that our method utilizes only the SDMap priors, while "S+H" indicates the utilization of the both. Our method consistently outperforms the baseline method under various weather conditions and in scenarios involving viewpoint occlusion.

In order to conduct a comprehensive evaluation of the effectiveness of our method across varying distances, we set three distinct perception ranges along the direction of vehicle travel: $[-30\text{m}, 30\text{m}]$, $[-60\text{m}, 60\text{m}]$, $[-120\text{m}, 120\text{m}]$. Additionally, we utilized different map resolutions, specifically 0.15m for the short range of $[-30\text{m}, 30\text{m}]$ and 0.3m for the rest two longer ranges. We use intersection-over-union (IoU) as the first metrics for segmentation results.

Furthermore, we incorporate a post-process to get the vectorized map and evaluate it using the average precision (AP). Following Dong et al. (2022), we set the threshold of IoU as 0.2 and threshold of CD as 0.2m, 0.5m, 1.0m.

## 5.2 IMPLEMENTATION DETAILS

P-MapNet is trained with the NVIDIA GeForce RTX 3090 GPU. We use the Adam (Kingma & Ba, 2014) optimizer and StepLR schedule for trainning with a learning rate of $5 \times 10^{-4}$. For fairness comparison, we adopt the EfficientNet-B0 (Tan & Le, 2019) pretrained on ImageNet (Russakovsky et al., 2015) as the perspective view image encoder and use a MLP to convert to the BEV features. To encode the point clouds for the LiDAR BEV features, we utilize the PointPillars framework (Lang et al., 2019), operating at a dimensionality of 128. During the pretraining phase for the HDMap

Table 1: **P-MapNet achieves state-of-the-art on NuScenes *val* set.** The symbol "†" denotes results reported in Xie et al. (2023), Dong et al. (2022), while "NMP" represents the "HDMapNet+NMP" configuration as described in Xiong et al. (2023). For superlong-range perception, we compared with SuperFusion Dong et al. (2022) and BEVFusion (Liu et al., 2022b). "C" and "L" respectively refer to the surround-view cameras and LiDAR inputs. Ours used both SDMap priors and HDMap priors.

| Range | Method | Modality | Divider | Ped Crossing | Boundary | mIoU |
|---|---|---|---|---|---|---|
| | VPN † | C | 36.5 | 15.8 | 35.6 | 29.3 |
| | Lift-Splat-Shoot † | C | 38.3 | 14.9 | 39.3 | 30.8 |
| | HDMapNet | C | 40.5 | 19.7 | 40.5 | 33.57 |
| $60 \times 30\,m$ | NMP † | C | 44.15 | 20.95 | **46.07** | 37.05 |
| | HDMapNet | C+L | 45.9 | 30.5 | 56.8 | 44.40 |
| | P-MapNet(Ours) | C | **44.3** | 23.3 | 43.8 | 37.13 |
| | P-MapNet(Ours) | C+L | **54.2** | **41.3** | **63.7** | **53.07** |
| | BEVFusion † | C+L | 33.9 | 18.8 | 38.8 | 30.5 |
| $90 \times 30\,m$ | SuperFusion † | C+L | 37.0 | 24.8 | 41.5 | 34.43 |
| | P-MapNet(Ours) | C+L | **44.73** | **31.03** | **45.47** | **40.64** |

Table 2: **Quantitative results of map segmentation mIoU scores (%).** Performance comparison of HDMapNet (Li et al., 2022) baseline and ours on the nuScenes val set (Caesar et al., 2020). In the case of using "HD.Prior." with P-MapNet, "Epoch" represents the number of refinement epochs.

| Range | Method | SD.Prior. | HD.Prior. | Modality | Epoch | Divider | Ped Crossing | Boundary | mIoU | FPS |
|---|---|---|---|---|---|---|---|---|---|---|
| | HDMapNet | | | C | 30 | 40.5 | 19.7 | 40.5 | 33.57 | 35.4 |
| | P-MapNet | ✓ | | C | 30 | 44.1 | 22.6 | 43.8 | 36.83 (+3.26) | 30.2 |
| $60 \times 30\,m$ | P-MapNet | ✓ | ✓ | C | 10 | 44.3 | 23.3 | 43.8 | 37.13 (+3.56) | 12.2 |
| | HDMapNet | | | C+L | 30 | 45.9 | 30.5 | 56.8 | 44.40 | 21.4 |
| | P-MapNet | ✓ | | C+L | 30 | 53.3 | 39.4 | 63.1 | 51.93 (+7.53) | 19.2 |
| | P-MapNet | ✓ | ✓ | C+L | 10 | **54.2** | **41.3** | **63.7** | **53.07 (+8.67)** | 9.6 |
| | HDMapNet | | | C | 30 | 39.2 | 23.0 | 39.1 | 33.77 | 34.2 |
| | P-MapNet | ✓ | | C | 30 | 44.8 | 30.6 | 45.6 | 40.33 (+6.56) | 28.7 |
| $120 \times 60\,m$ | P-MapNet | ✓ | ✓ | C | 10 | 45.5 | 30.9 | 46.2 | 40.87 (+7.10) | 12.1 |
| | HDMapNet | | | C+L | 30 | 53.2 | 36.9 | 57.1 | 49.07 | 21.2 |
| | P-MapNet | ✓ | | C+L | 30 | 63.6 | 50.2 | 66.8 | 60.20 (+11.13) | 18.7 |
| | P-MapNet | ✓ | ✓ | C+L | 10 | **65.3** | **52.0** | **68.0** | **61.77 (+12.70)** | 9.6 |
| | HDMapNet | | | C | 30 | 31.9 | 17.0 | 31.4 | 26.77 | 22.3 |
| | P-MapNet | ✓ | | C | 30 | 46.3 | 35.7 | 44.6 | 42.20 (+15.43) | 19.2 |
| $240 \times 60\,m$ | P-MapNet | ✓ | ✓ | C | 10 | 49.0 | 40.9 | 46.6 | 45.50 (+18.73) | 9.1 |
| | HDMapNet | | | C+L | 30 | 40.0 | 26.8 | 42.6 | 36.47 | 13.1 |
| | P-MapNet | ✓ | | C+L | 30 | 52.0 | 41.0 | 53.6 | 48.87 (+12.40) | 10.9 |
| | P-MapNet | ✓ | ✓ | C+L | 10 | **53.0** | **42.6** | **54.2** | **49.93 (+13.46)** | 6.6 |

prior, we trained for 20 epochs for each range. Subsequently, we combined the *BEV Feature Fusion* with the *HDMap Prior Refinement* module and conducted an additional 10 epochs of training to obtain the final HDMap predictions.

## 5.3 RESULTS

**Segmentation Results.** First, we conducted a comparative analysis of our approach with the current state-of-the-art (SOTA) approaches in both short-range ($60m \times 30m$) perception and long-range ($90m \times 30m$) with 0.15m resolution. As indicated in Tab. 1, our method exhibits superior performance compared to both existing vision-only and fused methods. Specifically, our method conducted experiments at 0.3m resolution for long-range perception, after which we upsample the predictions to 0.15m resolution and apply certain morphological operations to ensure a fair comparison. Secondly, we performed a performance comparison with HDMapNet (Li et al., 2022) at various distances and using different sensor modalities, with the results summarized in Tab. 2 and Tab. 3. Our method achieves a remarkable 13.4% improvement in mIOU at a range of $240m \times 60m$. It is noteworthy that the effectiveness of SD Map priors becomes more pronounced as the perception distance extends beyond or even surpasses the sensor detection range, thus validating the efficacy of SD Map priors. Lastly, our utilization of HD Map priors contributes to additional performance improvements by refining the initial prediction results to be more realistic and eliminating any anomalies, as demonstrated in Fig. 4. For additional qualitative experiment results, please refer to the supplementary materials in our Appendix C.

**Vectorization Results.** We also conducted a comparison of vectorization results by employing post-processing to obtain vectorized HD Maps. As detailed in Appendix B.4, we achieve the best instance detection AP results across all distance ranges. The visualization of the vectorized HD Map output can be found in Appendix C.

Table 3: **Quantitaive results of map segmentation on Argoverse2 *val* set.** We conducted a comparison between the P-MapNet method and HDMapNet (Li et al., 2022), using only surround-view cameras as input, and our results demonstrated superior performance.

| Range | Method | Divider | Ped Crossing | Boundary | mIoU |
|---|---|---|---|---|---|
| $60 \times 30\,m$ | HDMapNet | 41.7 | 19.2 | 30.8 | 30.56 |
| | P-MapNet (S) | 51.1 | 25.6 | 43.7 | 40.13 (+9.57) |
| | P-MapNet (S+H) | **51.5** | **26.4** | **43.7** | **40.53 (+9.97)** |
| $120 \times 60\,m$ | HDMapNet | 42.9 | 23.6 | 34.7 | 33.73 |
| | P-MapNet (S) | 48.6 | 32.0 | 45.8 | 42.13 (+8.40) |
| | P-MapNet (S+H) | **49.5** | **32.3** | **45.8** | **42.53 (+8.80)** |

Table 4: **Ablations about SD Maps fusion method.** The experiments are conducted with range of $120 \times 60m$ and C+L as inputs. "w/o SDMap" is the baseline method (Li et al., 2022). "w/o SDMap, w Self.Attn" only employed self-attention, and cross-attention, which integrates SD Map priors, was not utilized.

| Fusion Method | Divider | Ped Crossing | Boundary | mIoU |
|---|---|---|---|---|
| w/o SDMap | 53.2 | 36.9 | 57.1 | 49.07 |
| w/o SDMap, w/ Self.Attn. | 57.7 | 42.0 | 60.6 | 53.43 |
| Simply-concat | 59.4 | 43.2 | 61.6 | 54.73 |
| CNN-concat | 60.2 | 45.5 | 63.1 | 56.27 |
| Cross.Attn. | **63.6** | **50.2** | **66.8** | **60.20** |

## 5.4 ABLATION STUDY

All experiments are conducted on nuScenes val set with a perception range of $120m \times 60m$ and the camera-LiDAR fusion(C+L) configuration. More ablation studies are provided in Appendix A.3.

**Effectiveness of SDMap Prior Fusion.** To validate the effectiveness of our proposed fusion approach for SDMap priors, we experimented with various fusion strategies, the details of which are summarized in Tab. 4. In an initial evaluation, a straightforward concatenation (termed "Simple-concat") of the rasterized SDMap with BEV features $\mathcal{B}_{\text{sensor}}$ led to a mIoU boost of 4%. A more involved approach, where we deployed CNNs to encode and concatenate the rasterized SDMap, furthered this improvement to 6%. Nonetheless, the straightforward concatenation techniques were hampered by spatial misalignment issues, preventing the full capitalization of the SDMap priors' potential. Interestingly, leveraging self-attention solely for BEV queries also enhanced performance. Among all the approaches tested, our method anchored on cross-attention demonstrated the most substantial gains.

**Pretraining and Masking Strategy.** We conducted a comparison between results with and without pre-training, which clearly demonstrates that pretraining is effective in capturing HDMap priors. We devised two distinct mask strategies: the grid-based strategy and the random-mask strategy. Our approach, which utilized the random sampling strategy, produced the most promising results. For additional details, please refer to Appendix B.5.

## 6 DISCUSSION AND CONCLUSION

In summary, P-MapNet stands as a pioneering framework that harnesses prior information from both SDMap and HDMap, marking a significant advancement in this field. Our attention-based architecture has proven highly adaptable to the challenges posed by misaligned SDMap data. Additionally, the utilization of a masked autoencoder has allowed us to effectively capture the prior distribution of HDMap, facilitating its role as a refinement module for addressing concerns such as occlusions and artifacts. Extensive experimentation has demonstrated that our method serves as an effective *far-seeing* solution for HDMap construction and localization challenges in autonomous driving scenarios.

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

# Appendix

## A FURTHER STUDY ON SDMAP PRIOR

### A.1 INTEGRATING SDMAP PRIOR INTO END-TO-END VECTORIZED FRAMEWORK

As demonstrated in Tab.5, to confirm the universality of our SDMap prior, we integrated our SDMap Prior Module into MapTR(Liao et al., 2022) (with only minor modifications), a state-of-the-art end-to-end framework, referred to as the MapTR-SDMap method. Our MapTR-SDMap method also led to a significant improvement in mean Average Precision (mAP).

The visualization results in Fig. 5 also show that MapTR-SDMap performs better under the most challenging $240 \times 60 m$ ultra-long range of perception. It can also be seen that the segmentation-post-processing approach has stable results because it is sense-prediction, while the end-to-end vectorization approach still has some challenges such as significant predictive bias and challenges in keypoint selection. In conclusion, our proposed SDMap Prior fusion method demonstrates performance improvement in both the segmentation-postprocessing framework and the end-to-end framework.

Table 5: **Comparisons with MapTR (Liao et al., 2022) on nuScenes *val* set.** We conducted a comparison between MapTR fused with the SDMap prior method (MapTR-SDMap) and the vanilla MapTR (Liao et al., 2022), using only surround-view cameras as input and we use predefined CD thresholds of 0.5m, 1.0m and 1.5m. Our results demonstrated superior performance, highlighting the effectiveness of our SDMap prior fusion method.

| Range | Method | Divider | Ped Crossing | Boundary | mAP |
|---|---|---|---|---|---|
| $60 \times 30\,m$ | MapTR | 49.50 | 41.17 | 51.08 | 47.25 |
| | MapTR-SDMap | **50.92** | **43.71** | **53.49** | **49.37 (+2.21)** |
| | P-MapNet | 26.08 | 17.66 | 48.43 | 30.72 |
| $120 \times 60\,m$ | MapTR | 26.00 | 18.89 | 15.73 | 20.20 |
| | MapTR-SDMap | **27.23** | 21.95 | 19.50 | 22.89 (+2.69) |
| | P-MapNet | 19.50 | **24.72** | **42.48** | **28.90** |
| $240 \times 60\,m$ | MapTR | 12.69 | 7.17 | 4.23 | 8.03 |
| | MapTR-SDMap | **22.74** | 16.34 | 10.53 | 16.53 (+8.50) |
| | P-MapNet | 14.51 | **25.63** | **28.11** | **22.75** |

### A.2 INCONSISTENCIES BETWEEN GROUND TRUTH AND SDMAPS

**Influence of Inconsistencies between Ground Truth and SDMaps.** As detailed in Sec. 3, our SDMap priors are derived from OpenStreetMap (OSM). Nonetheless, due to discrepancies between labeled datasets and actual real-world scenarios, not all roads datasets are comprehensively annotated. This leads to incongruities between the SDMap and HDMap. Upon a closer examination of OSM, we noticed that there is a category in OSM called *service* road, which is for access roads leading to or located within an industrial estate, camp site, business park, car park, alleys, etc.

Incorporating *service* category roads can enrich the SDMap prior information with greater details. However, it also implies a potential increase in inconsistencies with dataset annotations. In light of this, we take ablation experiments to determine the advisability of incorporating *service* category roads.

As shown in Fig. 6, we select two cases to demonstrate the impact of inconsistencies between datasets map annotations and SDMaps. Specifically, in the Fig. 6(a), the inclusion of a *service* roads (an industrial estate) yields a positive outcome, where the SDMap aligns well with the ground truth dataset.

Nevertheless, in most cases, SDMaps with *service* roads are inconsistent with the ground truth of datasets, primarily due to the lack of detailed annotations for internal roads. Consequently, during the training process, the network learns the distribution of most service roads and filters them as noise. This inadvertently led to some primary roads being erroneously filtered out as noise. As depicted in Fig. 6(b), the service road (two alleys) highlighted in the red box is absent in the ground truth. The network perceives it as noise and consequently does not produce the corresponding road. However, it also neglects to generate a road for the primary route indicated in the ground truth, delineated within the green box, resulting in a significant discrepancy. Conversely, the network that

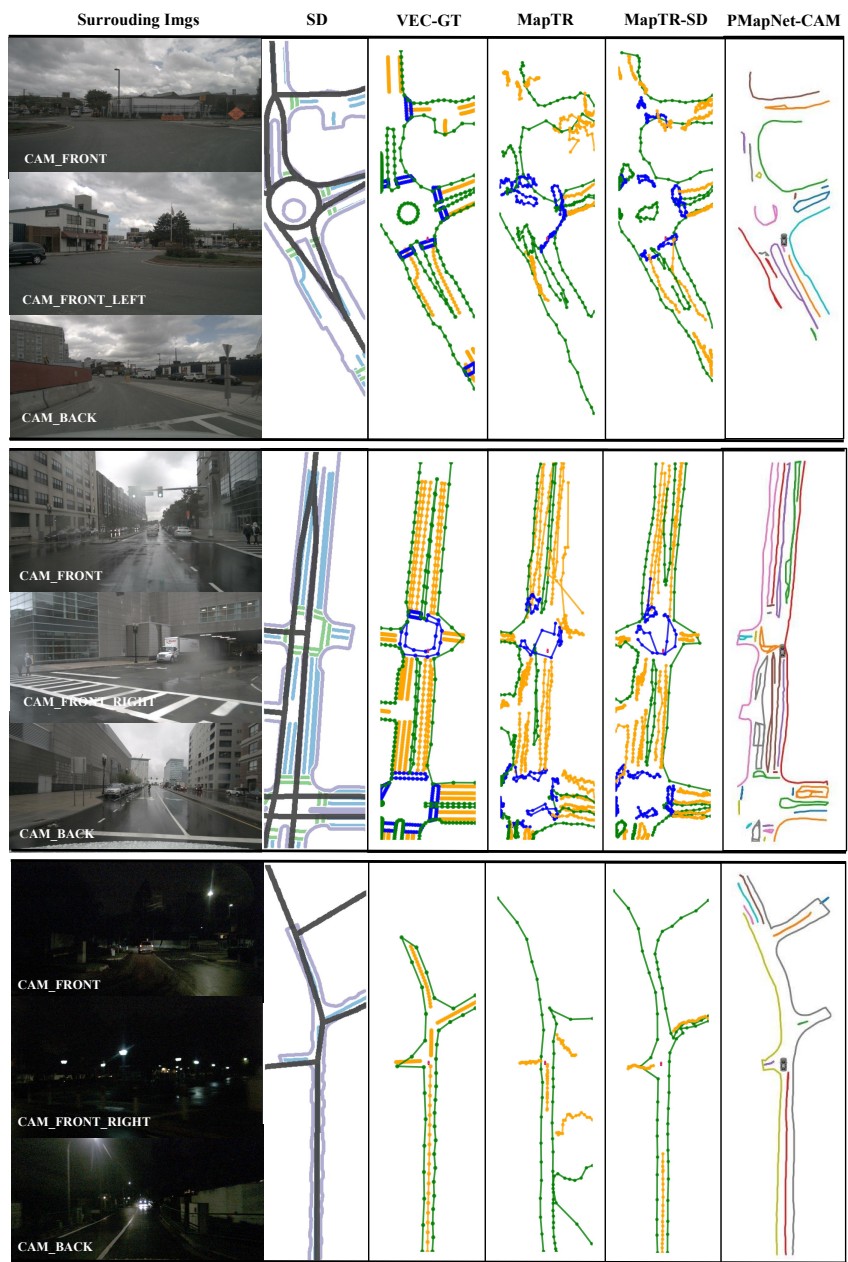

Figure 5: **The qualitative results of vectorized results on a perception range of** $240m \times 60m$ **.** We integrate the SDMap Prior Module into the MapTR(Liao et al., 2022) (with only minor modifications), referred to as the MapTR-SD. PMapNet-CAM is our method with both SDMap prior and HDMap prior utilizing the post process.

excludes *service* roads avoids learning numerous erroneous SDMap distributions. This enables the network to more effectively assimilate SDMap information pertaining to main roads, even though many detailed SDMaps may be missing. The visualization in the right side of Fig. 6(b) demonstrates that the SDMap prior effectively guides the generation of HDMap. It even reconstructs the pedestrian crosswalks and lane at remote intersections, even though these reconstructions do not align with the actual ground truth.

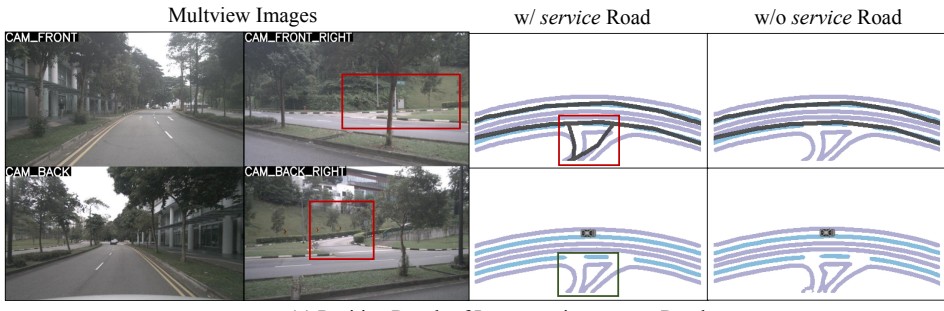

(a) Positive Result of Incorporating *service* Road

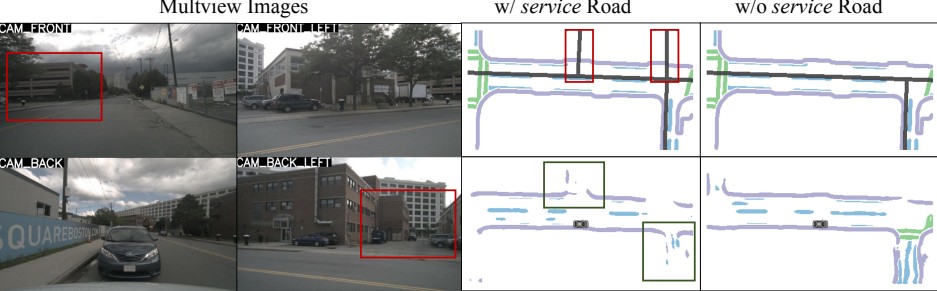

(b) Negative Result of Incorporating *service* Road

Figure 6: The two scenarios underscore the effects of discrepancies between the ground truth and SDMaps. (b) shows that the network filters most of the *service* roads as noise since majority of them are incongruous with the ground truth, which affects the performance of trunk roads. SDMap prior exhibits commendable efficacy when *service* roads are not introduced. (a) demonstrates that when the distribution of the *service* road deviates from the norm, its performance is enhanced since the network refrains from filtering it out as noise.

Table 6: **Quantitative results on different OSM category.** Incorporating *service* road introduce richer information but also involve inconsistency. In terms of segmentation mIoU results, the absence of *service* roads in the SDMap prior leads to an improvement of approximately 2% in performance.

| With *service* road | Divider | Ped Crossing | Boundary | mIoU |
|---|---|---|---|---|
| w/ *service* | 62.4 | 47.9 | 65.3 | 58.53 |
| w/o *service* | **63.6** | **50.2** | **66.8** | **60.20** |

In terms of quantitative metrics, as show in Tab. 6, excluding *service* roads results in a 2% mIoU improvement. The modest difference in these metrics suggests that the network can effectively filter out noise when introduced to numerous SDMaps that deviate from the ground truth. It further emphasizes the effectiveness of an SDMap focused on main roads in guiding the generation of the HDMap.

**Visualization Analysis of the Inconsistencies between Ground Truth and SDMaps.** As seen in Fig. 7, we select a case to show the negative result in the near side due to the inconsistencies. Obviously, the baseline shows that when SDMap prior information is not integrated, both the left and right forks at the near side can be predicted, but the far side cannot be predicted clearly due to weather reasons and visual distance.

When leveraging the SDMap prior to bolster HDMap generation, the predictions for the near side forks roads deteriorate due to the SDMap's exclusive focus on trunk roads. Furthermore, incorporating the HDMap prior alleviates artifacts and bridges the gaps, but this inadvertently diminishes prediction performance on the near side fork roads, as shown in Fig. 7(a).

However, we also validate the case using the model with *service* roads as in Fig. 7(b). The network perceives the *service* SDMap of the fork road (to the industrial park) as noise and filters it out,

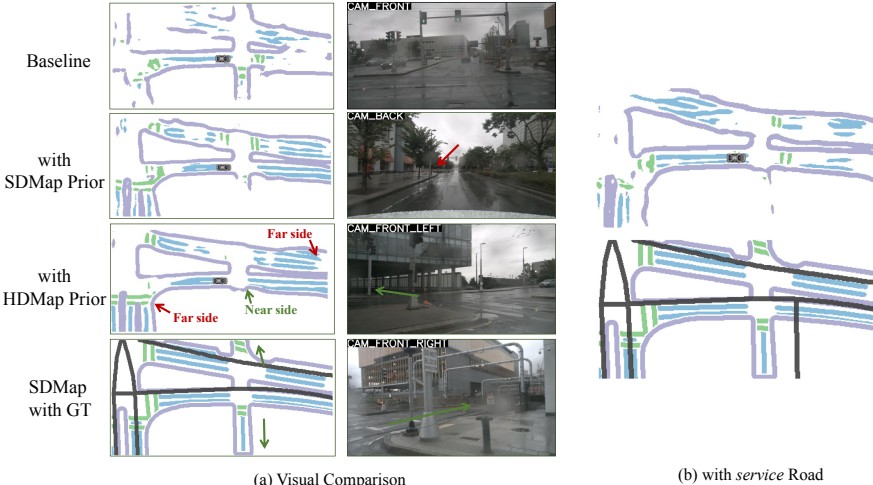

(a) Visual Comparison

(b) with *service* Road

Figure 7: **The negative results in the near side fork roads.** (a) demonstrates that the baseline exhibits proficient performance in predicting near side forks. Yet, due to the SDMap's exclusive focus on main roads, the prediction accuracy for near side forks diminishes upon integrating both the SDMap and HDMap prior information. (b) shows that even if *service* road information is added, the network will filter out this SDMap as noise.

as mentioned in the previous part. The other left side fork road is not contained in the *service*. Consequently, it performs suboptimal as well since the SDMaps is not detailed enough and the discrepancies between SDMaps and dataset's ground truth.

In summary, we introduce SDMap priors information and undertaken a comprehensive analysis of several intriguing cases. Our intent is to offer insights that may inspire future researchers to further the utilization of SDMap priors in HDMap generation.

### A.3   ABLATION OF ATTENTION LAYERS

As the number of transformer layers increases, performance of our method improves, but it eventually reaches a point of saturation since the SDMap priors contain low-dimensional information, excessively large network layers are susceptible to overfitting, as shown in Tab. 7.

Table 7: **Ablations about the number of Attention Layer.** During training, we evaluated memory usage with a batch size of 4, while for inference, we measured frames per second (FPS) with a batch size of 1.

| Attention Layer | Divider | Ped Crossing | Boundary | mIoU | Memory | FPS |
|---|---|---|---|---|---|---|
| 1 | 62.6 | 48.4 | 65.6 | 58.87 | 19.03 | 19.60 |
| 2 | **63.6** | **50.2** | **66.8** | **60.20** | 20.2 | 18.56 |
| 4 | 60.6 | 44.9 | 63.2 | 56.23 | 23.24 | 18.45 |
| 6 | 58.7 | 42.4 | 61.8 | 54.30 | OOM | - |

### A.4   ABLATION OF BEV FEATURE DOWNSAMPLING FACTOR

Different downsampling factor $d$ impact the size of feature map $\mathcal{B}_{small}$ in the fusion module. Larger feature maps convey more information but can result in higher GPU memory usage and slower inference speed. As shown in Tab. 8, to strike a balance between speed and accuracy, we opt for a size of $50 \times 25$.

Table 8: **Ablations about downsampling factor.** We conducted a comparison of mIOU results for feature sizes at a range of $120 \times 60m$ with different down-sampling multiples. The term "OOM" represents the GPU memory is exhausted. During training, we evaluated memory usage with a batch size of 4, while for inference, we measured frames per second (FPS) with a batch size of 1.

| Factor | Feature Map Size | Divider | Ped Crossing | Boundary | mIoU | Memory(GB) | FPS |
|--------|------------------|---------|--------------|----------|-------|------------|-------|
| $d = 2$ | $100 \times 50$ | - | - | - | - | OOM | - |
| $d = 4$ | $50 \times 25$ | **63.6** | **50.2** | **66.8** | **60.20** | 20.2 | 18.56 |
| $d = 8$ | $25 \times 12$ | 60.7 | 45.0 | 63.3 | 56.33 | 18.3 | 19.60 |

Table 9: **Ablations about mask proportion.** We use different random mask ratios for pre-training, with higher mask ratios being harder for the reconstruction.

| Mask Proportion | Divider | Ped Crossing | Boundary | mIoU |
|-----------------|---------|--------------|----------|-------|
| 25% | 64.8 | 51.4 | 67.6 | 61.27 |
| 50% | **65.3** | 52.0 | **68.0** | **61.77** |
| 75% | 64.7 | **52.1** | 67.7 | 61.50 |

## B  FURTHER STUDY ON HDMAP PRIOR

### B.1  PERCEPTUAL METRIC OF HDMAP PRIOR

The HDMap Priors Module endeavors to map the network output onto the distribution of HDMaps to make it appear more *realistic*. To evaluate the *realism* of the HDMap prior refinement Module output, we utilized a perceptual metric LPIPS(Zhang et al., 2018) (lower values indicate better performance). LPIPS leverages deep learning techniques to more closely simulate human visual perception differences, providing a more precise and human vision-aligned image quality assessment than traditional pixel-level or simple structural comparisons. The enhancements achieved in the HDMap Prior Module are considerably greater when compared to those in the SDMap Prior Module as demonstrated in Tab.10.

Table 10: **Perceptual Metric of HDmap Prior.** We utilizing the LPIPS metric to evaluate the *realism* of fusion model on $120m \times 60m$ perception range. And the improvements in the HDMap Prior Module are more significant compared to those in the SDMap Prior Module.

| Range | Method | Modality | mIoU↑ | LPIPS↓ | Modality | mIoU↑ | LPIPS↓ |
|-------|--------|----------|-------|--------|----------|-------|--------|
| $120 \times 60\,m$ | Baseline | C | 33.77 | 0.8050 | C+L | 49.07 | 0.7872 |
| | P-MapNet (S) | C | 40.33(+6.56) | 0.7926(1.54%)↓ | C+L | 60.20(+11.13) | 0.7607(3.37%)↓ |
| | P-MapNet (S+H) | C | **40.87(+7.10)** | **0.7717(4.14%)↓** | C+L | **61.77(+12.70)** | **0.7124(9.50%)↓** |
| $240 \times 60\,m$ | Baseline | C | 26.77 | 0.8484 | C+L | 36.47 | 0.8408 |
| | P-MapNet (S) | C | 42.20(+15.43) | 0.8192(3.44%)↓ | C+L | 48.87(+12.40) | 0.8097(3.70%)↓ |
| | P-MapNet (S+H) | C | **45.50(+18.73)** | **0.7906(6.81%)↓** | C+L | **49.93(+13.46)** | **0.7765(7.65%)↓** |

### B.2  GENERALIZABILITY OF PRE-TRAINING REFRINEMENT MODULE

In order to verify the generalizability of our HDMap Prior refinement module, we pre-train on Argoverse2 and nuScenes datasets respectively, and refine on nuScenes dataset and test the prediction results mIOU. The results are shown in the Tab. 11, and it can be seen that the model pre-trained on Argoverse2 is only $0.64\%$ mIOU lower than the pre-trained model on nuScenes, which can prove that our refinement module indeed captures the HDMap priors information with high generalization rather than overfitting on the dataset.

### B.3  MASK PROPORTION EXPERIMENT

As show in Tab. 9, we test the effect of using different mask ratios for pre-training on the refinement results, too high a mask ratio will lead to the lack of valid information and the actual refinement process of the input difference is large, too low a mask ratio can not force the network to capture the HDMap priors, we choose the optimal $50\%$ as the ratio of pre-training of our method.

Table 11: **Cross-Data experiment of HDMap Priors.** We pre-trained the HDMap Prior Module on the Argoverse2 and nuScenes datasets, respectively, and then tested it on the nuScenes val set, using a model of $120 \times 60m$, using Lidar and cameras as inputs.

| Pre-Train Dataset | Divider | Ped Crossing | Boundary | mIoU ↑ | LPIPS ↓ |
|---|---|---|---|---|---|
| Argoverse v2 | 64.5 | 51.3 | 67.6 | 61.13 (+0.93) | 0.7203 (8.49%)↓ |
| Nuscense | 65.3 | 52.0 | 68.0 | 61.77 (+1.57) | 0.7124 (9.50%)↓ |

### B.4 VECTORIZATION RESULTS BY POST-PROCESSING

We compared the vectorization results by incorporating post-processing to generate vectorized HD Maps. As outlined in Tab. 12, we attained the highest instance detection AP results across all distance ranges.

Table 12: **Quantitative results of AP scores.** Performance comparison of vectorize map instances on the nuScense val setCaesar et al. (2020). For the AP metric, we follow the approach of Li et al. (2022), and we use predefined CD thresholds of 0.5m, 1.0m and 1.5m.

| Range | Method | SD.Prior. | HD.Prior. | Modality | Epoch | Divider | Ped Crossing | Boundary | mAP |
|---|---|---|---|---|---|---|---|---|---|
| $60 \times 30\,m$ | HDMapNet | | | C | 30 | 27.68 | 10.26 | 45.19 | 27.71 |
| | P-MapNet | ✓ | | C | 30 | 32.11 | 11.33 | 48.67 | 30.70 (+2.99) |
| | P-MapNet | ✓ | ✓ | C | 10 | 26.08 | 17.66 | 48.43 | 30.72 (+3.01) |
| | HDMapNet | | | C+L | 30 | 29.46 | 13.89 | 54.07 | 32.47 |
| | P-MapNet | ✓ | | C+L | 30 | 36.56 | 20.06 | 60.31 | 38.98 (+6.51) |
| | P-MapNet | ✓ | ✓ | C+L | 10 | **37.81** | **24.96** | **60.90** | **41.22 (+8.75)** |
| $120 \times 60\,m$ | HDMapNet | | | C | 30 | 14.40 | 8.98 | 34.99 | 19.46 |
| | P-MapNet | ✓ | | C | 30 | 19.39 | 14.59 | 38.69 | 24.22 (+4.76) |
| | P-MapNet | ✓ | ✓ | C | 10 | 19.50 | 24.72 | 42.48 | 28.90 (+9.44) |
| | HDMapNet | | | C+L | 30 | 21.11 | 18.90 | 47.31 | 29.11 |
| | P-MapNet | ✓ | | C+L | 30 | 28.30 | 25.67 | 52.51 | 35.49 (+6.38) |
| | P-MapNet | ✓ | ✓ | C+L | 10 | **30.63** | **28.42** | **53.27** | **37.44 (+8.33)** |
| $240 \times 60\,m$ | HDMapNet | | | C | 30 | 7.37 | 5.09 | 21.59 | 11.35 |
| | P-MapNet | ✓ | | C | 30 | 10.86 | 12.74 | 25.52 | 16.38 (+5.03) |
| | P-MapNet | ✓ | ✓ | C | 10 | 14.51 | **25.63** | 28.11 | 22.75 (+11.40) |
| | HDMapNet | | | C+L | 30 | 11.29 | 11.40 | 29.05 | 17.25 |
| | P-MapNet | ✓ | | C+L | 30 | 17.87 | 20.00 | **35.89** | 24.59 (+7.34) |
| | P-MapNet | ✓ | ✓ | C+L | 10 | **21.47** | 24.14 | 34.23 | **26.61 (+9.36)** |

Table 13: **Detailed runtime**. We conducted tests on the time consumption of each component in P-MapNet at a range of $60 \times 120m$ on one RTX 3090 GPU.

| Component | Runtime (ms) | Proportion |
|---|---|---|
| Image backbone | 7.56 | 7.63% |
| View transformation | 3.25 | 3.28% |
| Lidar backbone | 17.60 | 17.76% |
| SDMap prior module | 4.40 | 4.45% |
| HDMap prior refinement | 66.12 | 66.87 % |
| Total | 98.87 | 100 % |

### B.5 ABLATION OF MASK STRATEGY

The grid-based strategy uses a patch size of $20 \times 20$ pixels and keeps one of every two patches. And the random-mask strategy selects one of the patch sizes from $20 \times 20$, $20 \times 40$, $25 \times 50$, or $40 \times 80$ with a 50% probability for masking. The visualization results are presented in Figure 8. With pre-training, the refinement module effectively learns the HDMap priors. As depicted in Tab.14, our approach employing the random sampling strategy yielded the most favorable results.

### B.6 DETAILED RUNTIME

In Tab. 13, we provide the detailed runtime of each component in P-MapNet with camera and lidar inputs.

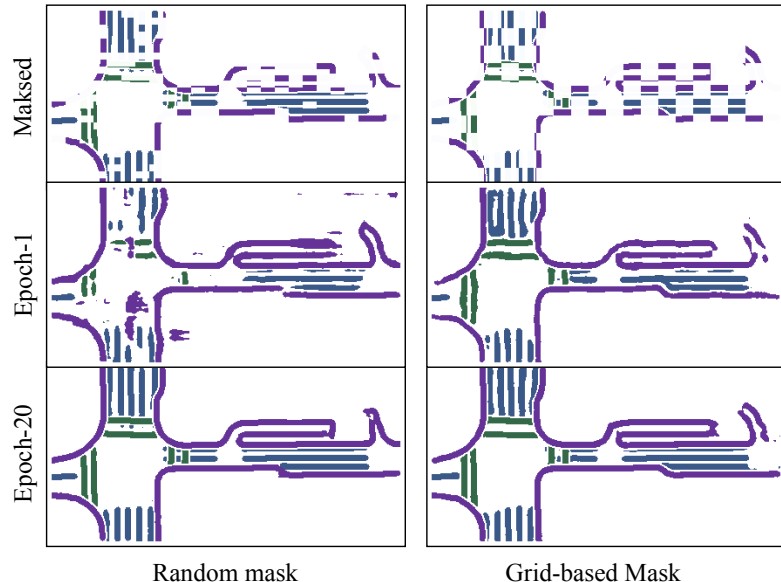

Figure 8: **Different mask strategies.** "Maksd" refers to the pre-training inputs after applying various masking strategies, and "Epoch-1" and "Epoch-20" denote the reconstruction results at the first and twentieth epochs of the pre-training process, respectively.

Table 14: **Ablations about mask strategy.** "w/o pretrain" signifies that we do not pre-train the HDMap Prior Refinement module. Interestingly, our random-mask method yields superior results in this context.

| Mask Strategy | Divider | Ped Crossing | Boundary | mIoU |
|---|---|---|---|---|
| w/o Pre-train | 64.1 | 51.4 | 67.4 | 60.97 |
| Gird-based | 65.1 | **52.3** | 67.8 | 61.73 |
| Random-mask | **65.3** | 52.0 | **68.0** | **61.77** |

# C  QUALITATIVE VISUALIZATION

## C.1  SEGMENTATION QUALITATIVE RESULTS

As depicted in the Fig. 10, Fig. 11, Fig.9 and Fig. 5, we provide additional perceptual results under diverse weather conditions, and our method exhibits superior performance.

## C.2  SD MAP DATA VISUALIZATION

We supplemented the SD Map data on both the Argoverse2 (Wilson et al., 2021) and nuScenes (Caesar et al., 2020) datasets, the specific details are outlined in Tab. 15. The visualization of SD map data and HD Map data facilitated by the dataset, is presented in Fig. 12 and Fig. 13.

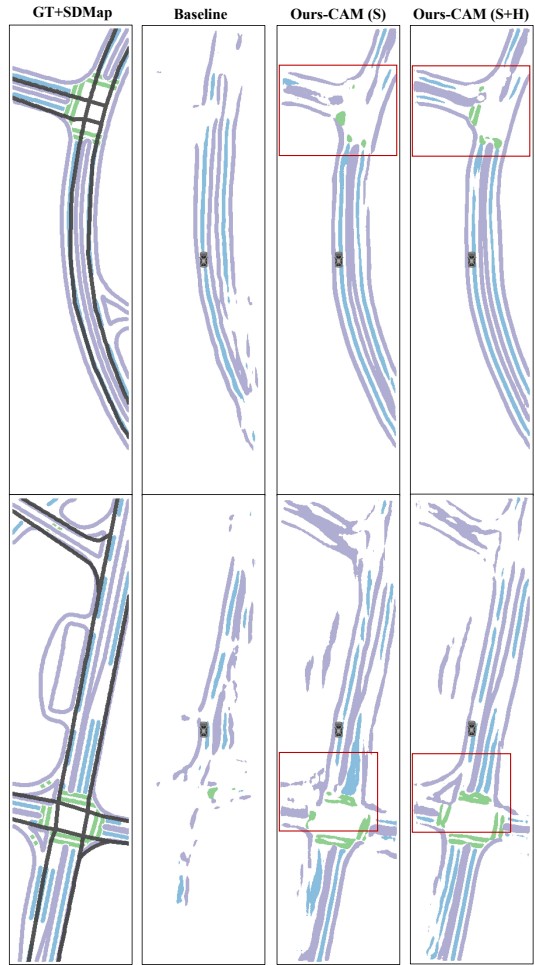

Figure 9: **The qualitative results of only camera method on a perception range of** $240m \times 60m$**.** The SDMap Prior Module improves the prediction results by fusing road structure priors. While HDMap Prior Module makes it closer to the distribution of HDMap to a certain extent, making it look more realistic.

Table 15: **SDMap data details.** In order to generate the SDMap data, we extract the *road*, *road link* and *special road* data from the *highway section* of OSM data and perform coordinate alignment and data filtering.

| Dataset | Sub-Map | Lane Numbers | Total Length(km) |
|---|---|---|---|
| NuScenes(Caesar et al., 2020) | Singapore-OneNorth | 576 | 23.4 |
| | Singapore-HollandVillage | 359 | 16.9 |
| | Singapore-Queenstown | 393 | 17.9 |
| | Boston-Seaport | 342 | 32.1 |
| Argoverse2(Wilson et al., 2021) | Austin | 193 | 46.5 |
| | Detroit | 853 | 160.6 |
| | Miami | 1226 | 178.2 |
| | Palo Alto | 315 | 33.4 |
| | Pittsburgh | 640 | 112.3 |
| | Washington DC | 1020 | 150.6 |

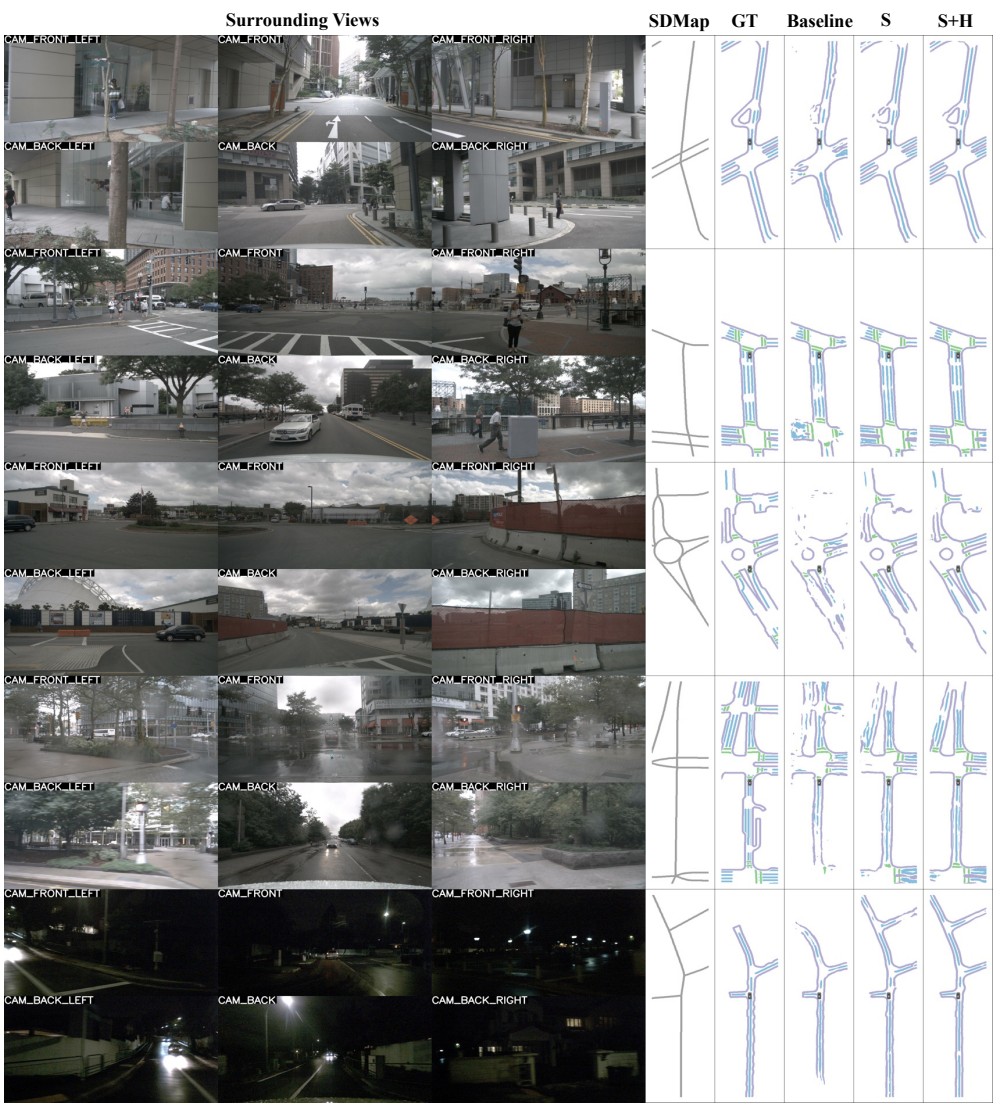

Figure 10: Visualization under various weather conditions was conducted, with the baseline method being HDMapNet (Li et al., 2022). The evaluation was performed using the C+L sensor configuration and a perception range of $240m \times 60m$. "S" indicates that our method utilizes only the SDMap priors, while "S+H" indicates the utilization of the both.

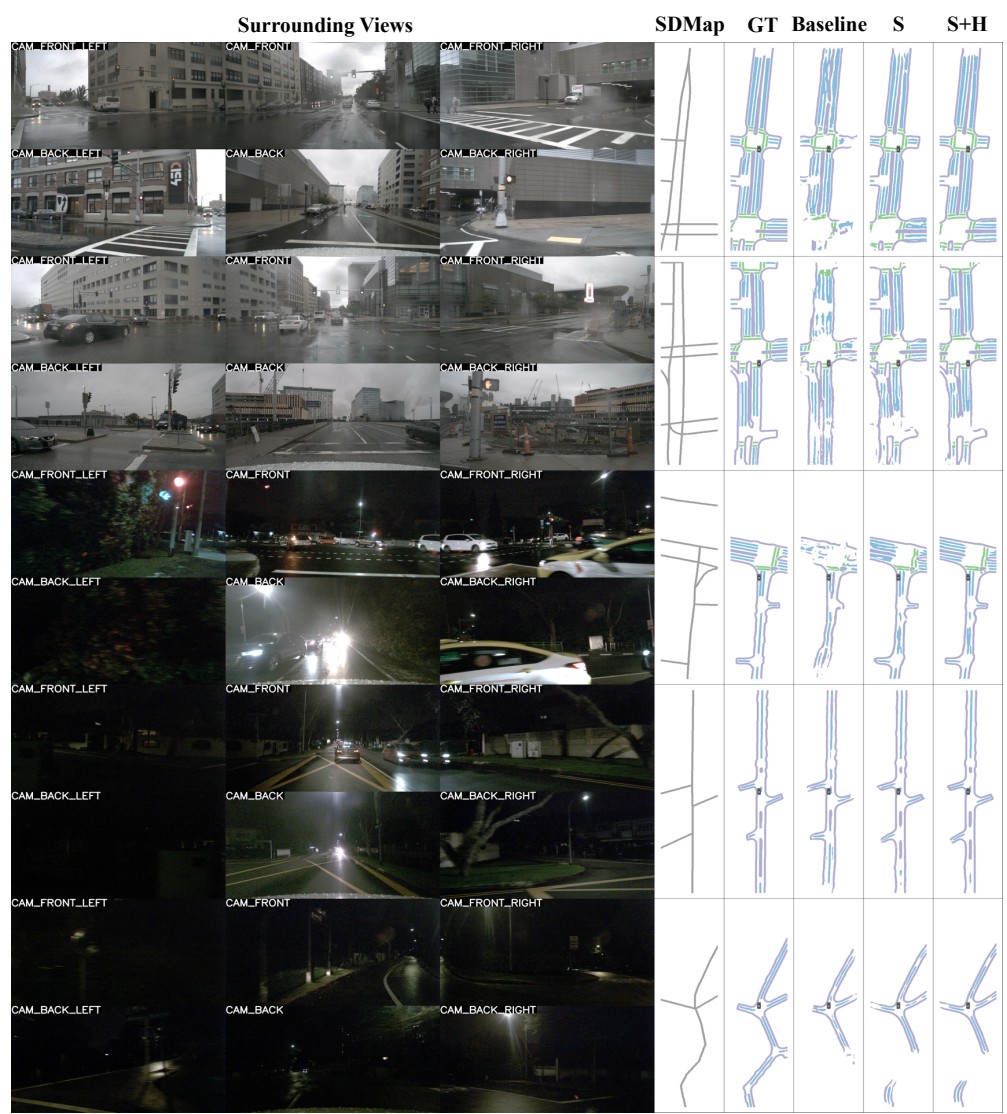

Figure 11: Visualization under various weather and light conditions was conducted, with the baseline method being HDMapNet (Li et al., 2022). The evaluation was performed using the C+L sensor configuration and a perception range of $240m \times 60m$. "S" indicates that our method utilizes only the SDMap priors, while "S+H" indicates the utilization of the both.

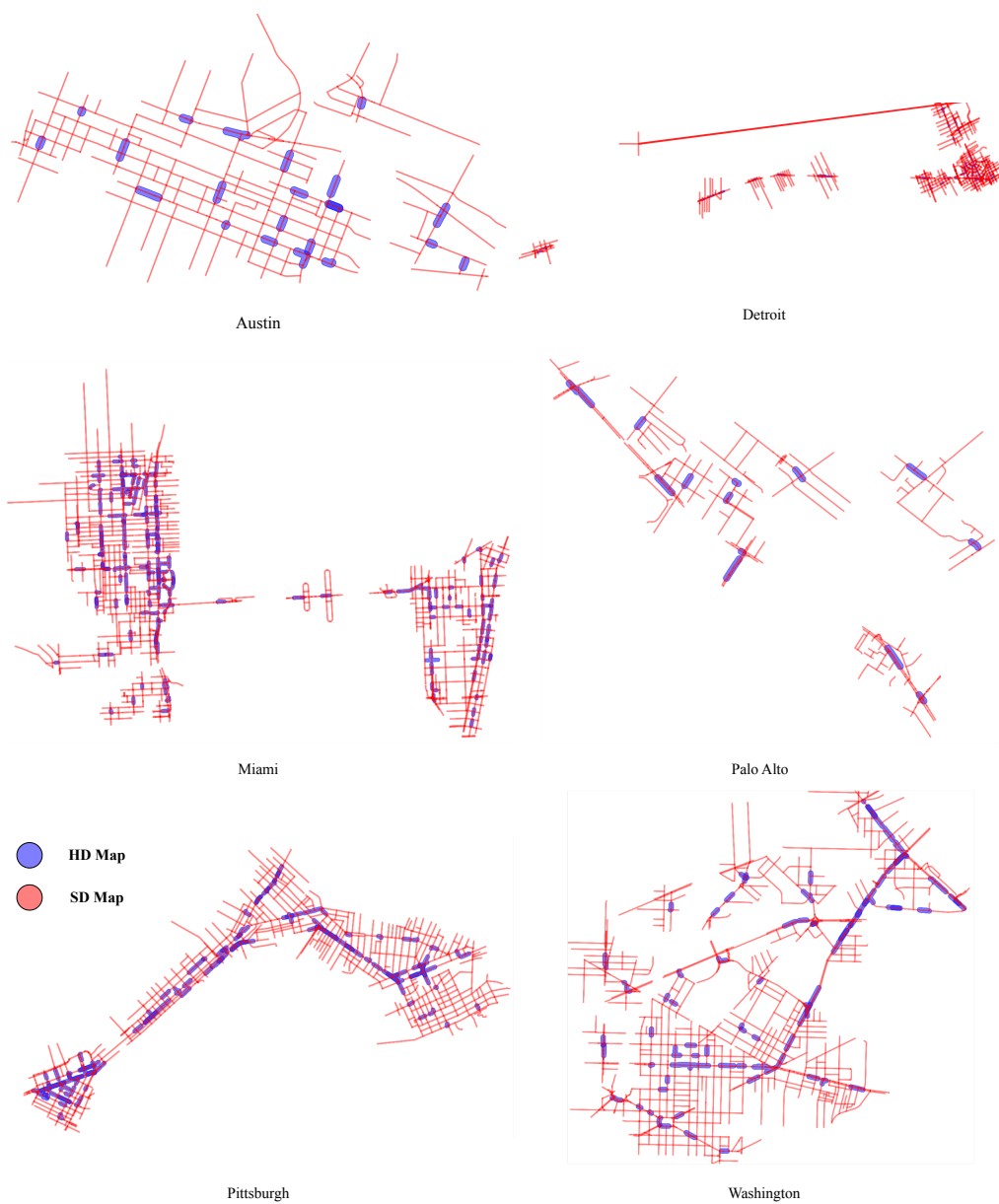

Figure 12: The visualizations of SD Map data and HD Map data on the Argoverse2 dataset.

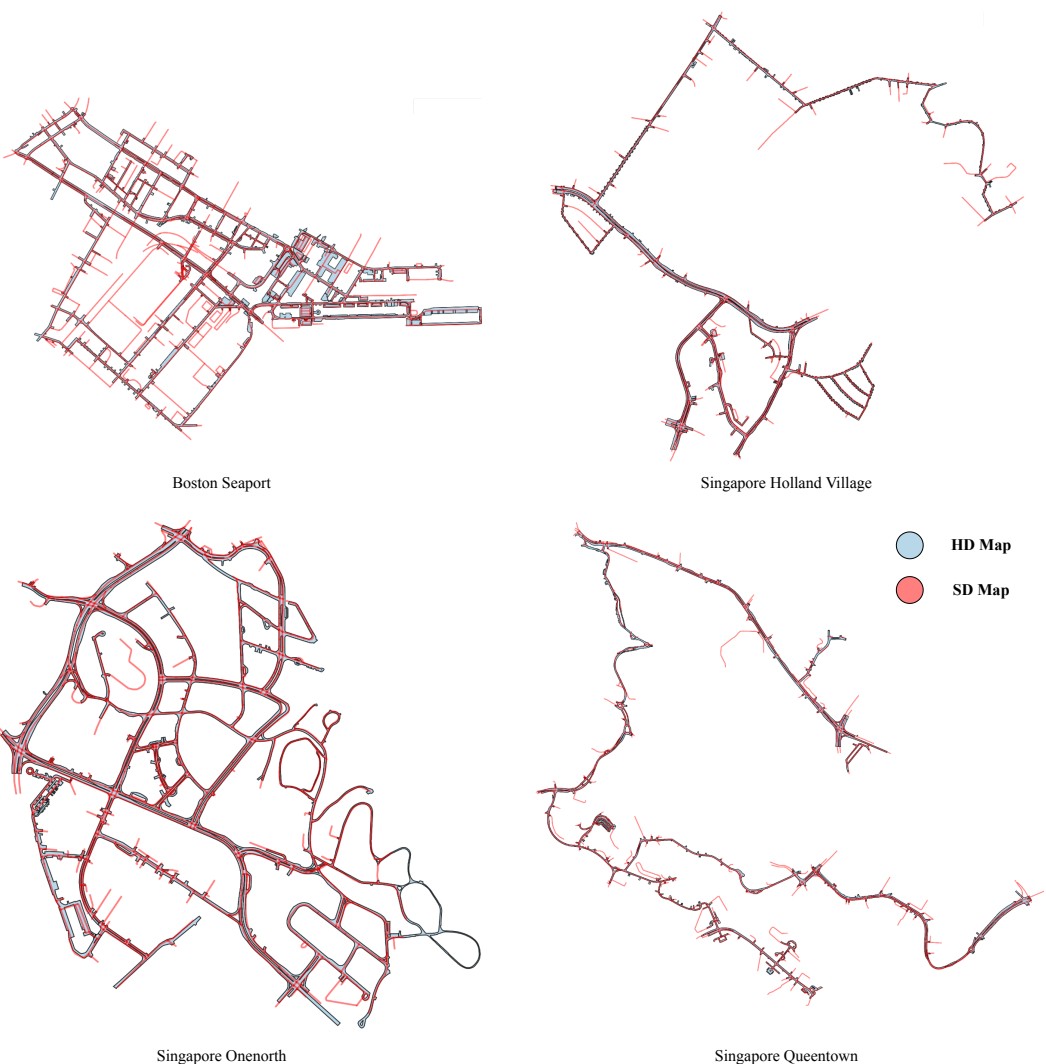

Figure 13: The visualizations of SD Map data and HD Map data on the nuScenes dataset.

