# OpenReview forum: "P-MapNet: Far-seeing Map Constructer Enhanced by both SDMap and HDMap Priors"
_ICLR.cc/2024/Conference — Submitted to ICLR 2024_

### Official Review · Reviewer_6mPa · 2023-10-30

**Soundness:** 2 fair
**Presentation:** 2 fair
**Contribution:** 2 fair
**Rating:** 5
**Confidence:** 4

**Summary:**

This paper presents a novel online rasterized HDMapping algorithm P-MapNet and focuses on exploiting priors in both SDMap and HDMap to get rid of the current reliance on expensive HDMaps for autonomous vehicles. The authors propose two novel designs within P-MapNet: a multi-head cross-attention-based SDMap prior module to settle the problem of SDMap misalignment and a ViT-style HDMap prior refinement module pre-trained on the masked-autoencoder methodology. In the experiments, P-MapNet is evaluated on both the NuScenes, where it achieves a 13.4% improvement in mIOU at the range of 240m * 60m, and Argoverse2 dataset, where it increases by 9.36 mAP compared to the baseline method demonstrating the effectiveness of its far-seeing solution for online HDMap construction and localization challenges in autonomous driving scenarios via both SDMap and HDMap priors.

**Strengths:**

1. This work provides a detailed explanation of a two-phase OSM data-based rasterized SDMap generation method, contributing to the advancement of research on SDMap utilization.

2. The main quantitative evaluations are performed on widely-used public datasets, namely NuScenes and Argoverse2, highlighting the salient performance improvement achieved by P-MapNet.

3. The proposed design is thoroughly evaluated through a comprehensive set of ablations investigating the SDMap fusion methods. These ablations demonstrate the design merits of the proposed approach.

**Weaknesses:**

1. The paper emphasizes the limitation of relying on HDMaps for autonomous vehicles to operate outside regions with this infrastructure, yet it relies heavily on HDMap priors to refine outputs and address issues such as broken and unnecessarily curved results. Additionally, the approach of generating HDMap with prior information from HDMap seems counterintuitive and unreasonable.

2. The paper falls short of providing the results under the setting of camera-only modality and combining both SDMap prior and HDMap prior modules. This omission raises concerns about the true impact of HDMap priors on the overall performance.

3. In terms of vectorization baseline results, the authors only reproduce HDMapNet under their new settings on the NuScenes dataset, without conducting a comparison with other state-of-the-art methods, both vectorized and rasterized-to-vectorized. This limits the thoroughness of the performance evaluation.

**Questions:**

1. Could you please explain the reasons for selecting rasterized representation and employing post-processing for vectorized results instead of directly using a vectorized network? In Section 2.1, you mentioned the limitations of methods relying solely on onboard sensors, but it is not clear how this relates to the chosen representation. Additionally, you mentioned that your network is designed in a BEV dense prediction manner and the structured output space of BEV HDMap cannot be guaranteed. Given these factors, why not use the vectorized representation directly, which naturally addresses the problem and eliminates the need for additional MAE pretraining methodology?

2. Can you provide a demonstration of why online generation of HDMap is necessary when given HDMap, and explain the relatively low increase in performance when HDMap priors are added?

3. Can you report the results under the camera-only modality and when both SDMap prior and HDMap prior modules are combined, to accurately reflect the genuine influence of HDMap priors?

4. Can you reproduce the results of more recent state-of-the-art methods in the new long-range settings to compare their performance with the vectorized results? This comparison should include, but not be limited to, VectorMapNet (mentioned but not compared with), MapTR, MapVR (which also perform rasterized-to-vectorized conversion), and PivotNet.

5. In relation to Table 2, could you explain the significant decrease in frames per second (FPS) and how this might impact downstream or practical applications?

6. Your SDMap Prior Module aligns misaligned SDMap with BEV features using multi-head cross-attention. However, what if there are localization errors present in the BEV features, which is a common occurrence in both camera-only and camera+lidar models?

Minor issues:
In your summary of contributions, "artefacts" should be corrected to "artifacts."
Regarding your summary of contributions, I am confused about the example of "P-MapNet is a far-seeing solution." Could you clarify what it is specifically used for?

---

> ### Author Response · Authors · 2023-11-16
>
> Dear reviewer, thank you for the professional feedbacks. We have updated the manuscript and marked the changes in red. Here are my answers to your related questions.
> ## Questions.1 & Questions.4 & Weaknesses.3:
>
> We further demonstrate the effectiveness of SD map prior by incorporating it into the state-of-the-art vectorized map constructor MapTR. The results are presented in **Appendix A.1** and **Fig.5**. It shows that the SD map prior can also improve the performance of MapTR on various ranges. Since the MAE HD map modelling method does not apply to MapTR, it is not evaluated.
>
> Meanwhile, we would like to point out that dense rasterized prediction on the BEV view still has advantages over direct vectorized map prediction methods. Vectorized representation is intrinsically troubled by control point selection ambiguity, which negatively impacts generalization in real-world autonomous driving applications. We believe in the future, rasterized map constructer and vectorized map constructor would co-exist in the academia and industry.
>
> | Range               | Method     | Divider | PedCrossing | Boundary | mAP                            |
> |---------------------|------------|---------|--------------|----------|--------------------------------|
> |    | MapTR      | 12.69   | 7.17         | 4.23     | 8.03                           |
> |  $240\times 60m$   | MapTR-SDMap | **22.74** |  16.34    | 10.53    | 16.53 (+8.50)            |
> |                     | P-MapNet   | 14.51   | **25.63**        | **28.11** |  **22.75**                          |
>
> ## Questions.2 & Weaknesses.1:
> We actually didn't have the HDMap, we just pre-train the mask auto-encoding in the train split dataset and then evaluate in the val split dataset. Furthermore, to verify the generalizability of the HDMap Prior Module, we conducted cross-data experiments as detailed in **Appendix B.2**. In these experiments, we pre-trained the model on the train split of the Argoverse2 dataset and then evaluated it on the validation data of nuScenes.This approach also yielded an improvement in the mIoU.
> Additionnally, a difference between our MAE and the conventional MAE may be the cause of this confusion. Specifically, a conventional MAE takes RGB image patches and directly regresses RGB image values of the whole image. Here, the input and output of our HDmap MAE are both rasterized segmentation masks. In our method, the output segmentation mask is supervised by a pixel-wise cross-entropy loss instead of the typical regression loss in a conventional MAE. As such our MAE can be naturally used for refinement during inference since the head is a segmentation head. To note, this segmentation mask MAE is exactly doing masked auto-encoding.
> ## Questions.3 & Weaknesses.2:
> We have updated the relevant experiments in **Tab.2**, and conducted relevant experiments in **Fig. 5** to obtain the visualization results and quantification results of the camera-only modality utilizing post-processing vectorization.
> ## Questions.5
> Regarding the issue of model runtime, we conducted further experiments to break down the inference time and discovered that the refinement module consumes a significant amount of time. In fact, the HDMap Prior Module is an optional component and can be chosen based on specific requirements. The detailed experimental results are presented in **Table 13**.
> | Component     | Runtime (ms)     | Proportion |
> |---------------------|------------|---------|
> |  Image backbone  | 7.56      | 7.63%   |
> |  View transformation | 3.25 | 3.28% |
> |  Lidar backbone | 17.60  |17.76%  |
> |SDMap prior module|4.40|4.45%|
> |HDMap prior module|66.12|66.87%|
> |Total |98.87| 100%|
> ## Questions.6
> However, localization errors only influence the SDMap obtained from the global SDMAP, the BEV feature is online generated from the onboard sensors.
> ## Minor issues:
> We have changed the spelling of the word. Online long-distance HDMap generation can obtain good priors for downstream planning tasks, thereby improving downstream planning and control tasks to be safer and smoother[1].
> ## Reference
> [1]Hao Dong, Xianjing Zhang, Xuan Jiang, Jun Zhang, Jintao Xu, Rui Ai, Weihao Gu, Huimin Lu, Juho Kannala, and Xieyuanli Chen. Superfusion: Multilevel lidar-camera fusion for long-range hd map generation and prediction. arXiv preprint arXiv:2211.15656, 2022

---

> > ### Comment · Reviewer_6mPa · 2023-11-21
> > **Feedback on the rebuttal materials**
> >
> > Thank you for the additional results. I acknowledge the utilization of SDMap, which is a neat idea and a promising direction. The explanation on the MAE part nearly almost my concern except one thing that what's the motivation of using a MAE to encode HDMap prior here. It add tremendous runtime cost according to your runtime analysis, while the improvement is relatively moderate as shown in Table 2 in the main paper.

---

> > > ### Author Response · Authors · 2023-11-23
> > >
> > > Dear reviewer, thank you very much for your further response. Here is our reply regarding the relevant issues.
> > > ## *Q: what's the motivation of using a MAE to encode HDMap prior here.*
> > >  A: Our MAE takes the rasterized HDMap as input and exploits a segmentation head to reconstruct the multi-class rasterized HDMap (supervised by cross-entropy). We choose this architecture as the tool for HDMap prior modelling because: (1) It can be trained in a self-supervised manner; (2) During inference, it can be naturally used for map refinement.
> > > There exist other generative modelling methods to capture the distribution of HDMaps, like generative adversarial networks (GAN) used in [1]. However, according to [1]'s time profiling, it can only achieve around 0.6 fps, showing the challenge of using GAN for HDMap refinement.
> > >
> > >
> > > ## *Q: It add tremendous runtime cost according to your runtime analysis, while the improvement is relatively moderate as shown in Table 2 in the main paper.*
> > >  A: As shown in Table.10 in the paper and below, the HDMap refinement module can significantly improve the LPIPS metric (which corresponds to map realism), despite that its impact on mIOU is relatively minor. This opinion is also echoed in a recent work [1].
> > > ****
> > > Tab.10 **Perceptual Metric of HDmap Prior.**
> > > | Range            | Method          | Modality | mIoU↑   | LPIPS↓        | Modality | mIoU↑   | LPIPS↓        |
> > > |------------------|-----------------|----------|---------|---------------|----------|---------|---------------|
> > > | $120 \times 60m$ | Baseline        | C        | 33.77   | 0.8050        | C+L      | 49.07   | 0.7872        |
> > > |                  | P-MapNet (S)    | C        | 40.33   | 0.7926 (1.54%)| C+L      | 60.20   | 0.7607 (3.37%)|
> > > |                  | P-MapNet (S+H)  | C        | **40.87**| **0.7717** (4.14%)| C+L  | **61.77**| **0.7124** (9.50%)|
> > > | $240 \times 60m$ | Baseline        | C        | 26.77   | 0.8484        | C+L      | 36.47   | 0.8408        |
> > > |                  | P-MapNet (S)    | C        | 42.20   | 0.8192 (3.44%)| C+L      | 48.87   | 0.8097 (3.70%)|
> > > |                  | P-MapNet (S+H)  | C        | **45.50**| **0.7906** (6.81%)| C+L  | **49.93**| **0.7765** (7.65%)|
> > > ****
> > > Meanwhile, HDMap refinement is an optional module because it works as a post-processing step. If practitioners pursue real-time performance, this module can discarded. If they have access to redandent computing resources (e.g., cloud-based automatic labeling systems), this module can be activated to pursue HDMap results that are more aligned with the realistic HDMap distribution.
> > >
> > > ## Reference
> > > [1]Zhu, Xiyue, et al. "MapPrior: Bird's-Eye View Map Layout Estimation with Generative Models." Proceedings of the IEEE/CVF International Conference on Computer Vision. 2023.

---

### Official Review · Reviewer_mZgt · 2023-11-01

**Soundness:** 3 good
**Presentation:** 3 good
**Contribution:** 3 good
**Rating:** 5
**Confidence:** 5

**Summary:**

This paper proposes a new approach called P-MapNet for far-seeing HD-Map generation. The proposed P-MapNet exploits the priors from SD-Map and HD-Map for long-distance HD-Map. This paper first generates SD-Map for nuScenes and Argoverse datasets and then presents the P-MapNet framework which is based on BEV and contains the SD-Map prior module and the HD-Map prior module. The HD maps are predicted by the segmentation head. The experiments can show the proposed method is effective, especially for long-range HD map prediction.

**Strengths:**

1. This paper presents a new HD map framework named P-MapNet aims for long-range HD map construction.
2. This paper builds the SD map for two datasets based on OpenStreetMap.
3. The proposed framework P-MapNet adopts the coarse SD-map prior and the fine-grained HD-map prior for far-seeing map construction.
4. The proposed P-MapNet obtains significant results compared to HDMapNet.

**Weaknesses:**

1. The experiments lack the comparisons with recent works, e.g., [1][2][3]. The baseline of HDMapNet is too old and weak.
2. The idea of building long-distance HD maps has been explored in previous works [4,5], the authors should clearly state the difference and the superiority of the proposed framework.
3. The proposed approach involves a large computation burden and  has lower inference speeds
4. The authors evaluate the proposed framework on the benchmark with the max range of 240x60 while I'm concerned about the superiority of the proposed framework compared to the methods trained with the same range. In addition, I'm concerned about whether the proposed method has a limited range, and what the range is when SD maps do not work.
5. The experiments about the downsampling factors of the SD map.
6. Experimental results about P-MapNet(S+H) without lidar.
7. In Sec.4.3, it's unclear how the initial maps are used in the mask image modelling and how the maps are refined during inference.

[1] Liao et.al. MapTR: Structured Modeling and Learning for Online Vectorized HD Map Construction. ICLR 2023.
[2] Liu et.al. VectorMapNet: End-to-end Vectorized HD Map Learning. ICML 2023.
[3] Ding et.al. PivotNet: Vectorized Pivot Learning for End-to-end HD Map Construction. ICCV 2023.
[4] Xiong et.al. Neural Map Prior for Autonomous Driving. CVPR 2023.

**Questions:**

1. I'm concerned about whether the proposed framework can be applied to vectorized methods, such as MapTR[1], though this paper tries to vectorize the map through post-processing.

[1] Liao et.al. MapTR: Structured Modeling and Learning for Online Vectorized HD Map Construction. ICLR 2023.

---

> ### Author Response · Authors · 2023-11-16
>
> Dear reviewer, thank you for the professional feedbacks. We have updated the manuscript and marked the changes in red. Here are my answers to your related questions.
> ## Weaknesses.1 & Weaknesses.4 & Questions.1:
>
> We further demonstrate the effectiveness of SD map prior by incorporating it into the state-of-the-art vectorized map constructor MapTR. The results are presented in **Appendix A.1** and **Fig.5**. It shows that the SD map prior can also improve the performance of MapTR on various ranges. Since the MAE HD map modelling method does not apply to MapTR, it is not evaluated.
>
> Meanwhile, we would like to point out that dense rasterized prediction on the BEV view still has advantages over direct vectorized map prediction methods. Vectorized representation is intrinsically troubled by control point selection ambiguity, which negatively impacts generalization in real-world autonomous driving applications. We believe in the future, rasterized map constructer and vectorized map constructor would co-exist in the academia and industry.
>
> Regarding the effective distance of SDMap, due to GPU memory constraints, we only supported testing up to 240 meters. Theoretically, SDMap has priors at all distances, and in ranges beyond sensor perception, SDMap alone provides valuable information. The model could potentially guess the form of the HDMap based on SDMap, although the generated HDMap may not be very accurate due to misalignment issues.
> | Range               | Method     | Divider | PedCrossing | Boundary | mAP                            |
> |---------------------|------------|---------|--------------|----------|--------------------------------|
> |    | MapTR      | 12.69   | 7.17         | 4.23     | 8.03                           |
> |  $240\times 60m$   | MapTR-SDMap | **22.74** |  16.34    | 10.53    | 16.53 (+8.50)            |
> |                     | P-MapNet   | 14.51   | **25.63**        | **28.11** |  **22.75**                          |
>
> ## Weaknesses.2:
> Our approach fundamentally differs from the method used in Neural Map Prior[1]. In [1], a Neural Map must be pre-constructed to enhance the online generation of HDMaps, which is then updated as the vehicle repeatedly traverses the same area, aiming to obtain a more accurate HDMap. Additionally, the furthest distance covered in their paper is 160m by 100m.
> Conversely, our method, based on the SDMap Prior Module, predicts HDMaps at ultra-long distances online, without the need for a prior traversal of the area. Furthermore, the maximum distance in our results extends to 240m by 60m.
> ## Weaknesses.3
> Regarding the issue of model runtime, we conducted further experiments to break down the inference time and discovered that the refinement module consumes a significant amount of time. In fact, the HDMap Prior Module is an optional component and can be chosen based on specific requirements. The detailed experimental results are presented in **Table 13**.
> | Component     | Runtime (ms)     | Proportion |
> |---------------------|------------|---------|
> |  Image backbone  | 7.56      | 7.63%   |
> |  View transformation | 3.25 | 3.28% |
> |  Lidar backbone | 17.60  |17.76%  |
> |SDMap prior module|4.40|4.45%|
> |HDMap prior module|66.12|66.87%|
> |Total |98.87| 100%|
> ## Weaknesses.5
> We have completed relevant experiments, see Appendix A.4 for details. SDMap downsampling factors are the same as factor **d**.
> ## Weaknesses.6
> We have updated relevant experiments in **Tab.2**
> ## Weaknesses.7
> A difference between our MAE and the conventional MAE may be the cause of this confusion. Specifically, a conventional MAE takes RGB image patches and directly regresses RGB image values of the whole image. Here, the input and output of our HDmap MAE are both rasterized segmentation masks. In our method, the output segmentation mask is supervised by a pixel-wise cross-entropy loss instead of the typical regression loss in a conventional MAE. As such our MAE can be naturally used for refinement during inference since the head is a segmentation head. To note, this segmentation mask MAE is exactly doing masked auto-encoding.
>
> We have further refined the description and ablation experiments of the HDMap Prior Module, as detailed in **Section 4.3** and **Appendix B**.
>
> ## Reference
> [1] Xiong et.al. Neural Map Prior for Autonomous Driving. CVPR 2023.

---

### Official Review · Reviewer_8h86 · 2023-11-03

**Soundness:** 2 fair
**Presentation:** 3 good
**Contribution:** 2 fair
**Rating:** 5
**Confidence:** 5

**Summary:**

This draft improves the accuracy of the online map construction task by introducing prior information from SDMap and HDMap. It uses surround images and point cloud data as input to obtain BEV (Bird's Eye View) features, and then utilizes attention mechanism to extract corresponding features from SDMap to generate better bev feature. It further ensures the continuity of segmentation results by using a pre-trained HDMap model based on MAE. The utilization of these two priors significantly enhances the accuracy of map construction.

**Strengths:**

1. The introduction of prior information of SDMap significantly improves the map accuracy at both short range and long range.
2. The highlight is the use of pretraining model based on MAE to ensure the continuity of segmentation result.
3. The ablation experiments in this article are quite comprehensive.

**Weaknesses:**

1. If a vectorization modeling approach is used, there might not be such discontinuities in results. This article should conduct further experiments to validate this matter.
2. The metrics of vectorization results should be compared with vectorized modeling methods.
3. The mask proportion of MAE should undergo some ablation experiments.
4. The benefits brought by the prior information of HDMap are too small.
5. There are some data inaccuracies in mIoU in Table 2.

**Questions:**

What data should be used to train this MAE-based ViT?  And and would the model overfit if all the data is utilized?

---

> ### Author Response · Authors · 2023-11-16
>
> Dear reviewer, thank you for the professional feedbacks. We have updated the manuscript and marked the changes in red. Here are my answers to your related questions.
> ## Weaknesses.1 & Weaknesses.2
> We further demonstrate the effectiveness of SD map prior by incorporating it into the state-of-the-art vectorized map constructor MapTR. The results are presented in **Appendix A.1** and **Fig.5**. It shows that the SD map prior can also improve the performance of MapTR on various ranges. Since the MAE HD map modelling method does not apply to MapTR, it is not evaluated.
>
> Meanwhile, we would like to point out that dense rasterized prediction on the BEV view still has advantages over direct vectorized map prediction methods. Vectorized representation is intrinsically troubled by control point selection ambiguity, which negatively impacts generalization in real-world autonomous driving applications. We believe in the future, rasterized map constructer and vectorized map constructor would co-exist in the academia and industry.
> | Range               | Method     | Divider | PedCrossing | Boundary | mAP                            |
> |---------------------|------------|---------|--------------|----------|--------------------------------|
> |    | MapTR      | 12.69   | 7.17         | 4.23     | 8.03                           |
> |  $240\times 60m$   | MapTR-SDMap | **22.74** |  16.34    | 10.53    | 16.53 (+8.50)            |
> |                     | P-MapNet   | 14.51   | **25.63**        | **28.11** |  **22.75**                          |
> ## Weaknesses.3
> We have added relevant ablation experiments. Specifically, in **Appendix B.5**, we conducted 25%, 50%, and 75% experiments. Too high a mask ratio will lead to the lack of valid information and the actual refinement process of the input difference is large, too low a mask ratio can not force the network to capture the HDMap priors。
> | Mask Proportion  | Divider | PedCrossing | Boundary | mAP                            |
> |--------------|---------|--------------|----------|--------------------------------|
> |    25%  | 64.8   | 51.4        |67.6     | 61.27                           |
> |    50% | **65.3** | 52.0    | **68.0**  | **61.77**             |
> |    75%   | 64.7   | **52.1**        | 67.7 | 61.50                         |
> ## Weaknesses.4
> This issue is also present in the work of MapPrior[1]. Essentially, our goal is to map the output of the SDMap Prior Module to the space of HDMaps. However, this process might not map onto the ground truth with high precision, hence the modest improvement in the mean Intersection over Union (mIoU) metric, which measures accuracy.In order to evaluate the realism after refining, we use a type of Perceptual Metrics, specifically LPIPS [2], to do so, and the results show that our HDMap Prior module has a significant improvement on predicting after refining than without refining. As detailed in **Appendix B.1**.
> | Method          | mIoU↑  | LPIPS↓ |
> |-----------------|--------|--------|
> | Baseline        | 49.07  | 0.7872 |
> | P-MapNet (S)    | 60.20  | 0.7607 |
> | P-MapNet (S+H)  | **61.77**  | **0.7124**|
> ## Weaknesses.5
> It may be a problem with our table layout. You may have misread the results of only cam and fusion model. We have revised the table **Tab.2**.
>
> ## Questions.1:
> Initially, we pre-trained our model using the train split of the nuScenes dataset, followed by evaluations on the validation split. This approach might have led to some overfitting. To verify the generalizability of the HDMap Prior Module, we conducted cross-data experiments as detailed in **Appendix B.2**. In these experiments, we pre-trained the model on the train split of the Argoverse2 dataset and then evaluated it on the validation data of nuScenes.This approach also yielded an improvement in the mIoU.
>
> ## Reference
> [1]Xiyue Zhu, Vlas Zyrianov, Zhijian Liu, and Shenlong Wang. Mapprior: Bird’s-eye view map layout estimation with generative models. arXiv preprint arXiv:2308.12963, 2023.
> [2]Richard Zhang, Phillip Isola, Alexei A Efros, Eli Shechtman, and Oliver Wang. The unreasonable effectiveness of deep features as a perceptual metric. In Proceedings of the IEEE conference on computer vision and pattern recognition, pp. 586–595, 2018.

---

### Official Review · Reviewer_Je8q · 2023-11-06

**Soundness:** 2 fair
**Presentation:** 3 good
**Contribution:** 2 fair
**Rating:** 5
**Confidence:** 4

**Summary:**

This paper aims to incorporate map priors, including priors both in SDMap and HDMap, to improve the performance of HDMap generation. Weakly aligned SDMap priors are extracted and encoded as an alternative conditioning branch. A masked autoencoder pretraining on nuscenes is utilized to refine the HDMap. Extensive experiments demonstrate the effectiveness of propose method.

**Strengths:**

1.	The paper is well structured. The presentation is clear and easy to understand.
2.	The novelty of the paper is good, The designs are motivated well and intuitive is good.
3.	MAE was used to improve the performance of map construction.
4.	The experiments are extensive, although some necessary experiments are missing.

**Weaknesses:**

1.	The benchmark is not compared reasonable. Comparison with recent advance works is needed.
2.	Some parts of the proposed method is not clarified clearly, such as the HDMap Refinement Module.
3.	The performance of run time is not competitive.
4.	Utilizing pretrained MAE as second-stage refinement is interesting. However, its generalization as a pretrained model is more worthy of exploration.

**Questions:**

1.	It is reasonable to compare with recent works with advance performance (e.g. [1], [2])
2.	Some parts of the proposed method is not clarified clearly, such as the HDMap Refinement Module. Please introduce more details about it. How can it refine the initial predictions with absent sidewalks and broken lane lines? Are there any insights?
3.	The performance of run time seems not competitive. Please explain about that.
4.	Utilizing pretrained MAE as second-stage refinement is interesting. However, its generalization as a pretrained model is more worthy of exploration. The reviewer wonder how it works when it come to other dataset, e.g., pretrained on nuscenes while inferenced on Ago.
Please explain my concerns and modify the manuscript according to the negatives. If all my concerns are well addressed, I will raise my score.
[1] Liao B, Chen S, Wang X, et al. Maptr: Structured modeling and learning for online vectorized hd map construction[J]. arXiv preprint arXiv:2208.14437, 2022.
[2] Liao B, Chen S, Zhang Y, et al. Maptrv2: An end-to-end framework for online vectorized hd map construction[J]. arXiv preprint arXiv:2308.05736, 2023.

---

> ### Author Response · Authors · 2023-11-16
>
> Dear reviewer, thank you for the professional feedbacks. We have updated the manuscript and marked the changes in red. Here are my answers to your related questions.
> ## Weaknesses.1 & Questions.1:
>
> We further demonstrate the effectiveness of SD map prior by incorporating it into the state-of-the-art vectorized map constructor MapTR. The results are presented in **Appendix A.1** and **Fig.5**. It shows that the SD map prior can also improve the performance of MapTR on various ranges. Since the MAE HD map modelling method does not apply to MapTR, it is not evaluated.
>
> Meanwhile, we would like to point out that dense rasterized prediction on the BEV view still has advantages over direct vectorized map prediction methods. Vectorized representation is intrinsically troubled by control point selection ambiguity, which negatively impacts generalization in real-world autonomous driving applications. We believe in the future, rasterized map constructer and vectorized map constructor would co-exist in the academia and industry.
> | Range               | Method     | Divider | PedCrossing | Boundary | mAP                            |
> |---------------------|------------|---------|--------------|----------|--------------------------------|
> |    | MapTR      | 12.69   | 7.17         | 4.23     | 8.03                           |
> |  $240\times 60m$   | MapTR-SDMap | **22.74** |  16.34    | 10.53    | 16.53 (+8.50)            |
> |                     | P-MapNet   | 14.51   | **25.63**        | **28.11** |  **22.75**                          |
>
> ## Weaknesses.2 & Questions.2:
> A difference between our MAE and the conventional MAE may be the cause of this confusion. Specifically, a conventional MAE takes RGB image patches and directly regresses RGB image values of the whole image. Here, the input and output of our HDmap MAE are both rasterized segmentation masks. In our method, the output segmentation mask is supervised by a pixel-wise cross-entropy loss instead of the typical regression loss in a conventional MAE. As such our MAE can be naturally used for refinement during inference since the head is a segmentation head. To note, this segmentation mask MAE is exactly doing masked auto-encoding.
>
> We have further refined the description and ablation experiments of the HDMap Prior Module, as detailed in **Section 4.3** and **Appendix B**.
>
> ## Weaknesses.3 & Questions.3:
> Regarding the issue of model runtime, we conducted further experiments to break down the inference time and discovered that the refinement module consumes a significant amount of time. In fact, the HDMap Prior Module is an optional component and can be chosen based on specific requirements. The detailed experimental results are presented in **Table 13**.
> | Component     | Runtime (ms)     | Proportion |
> |---------------------|------------|---------|
> |  Image backbone  | 7.56      | 7.63%   |
> |  View transformation | 3.25 | 3.28% |
> |  Lidar backbone | 17.60  |17.76%  |
> |SDMap prior module|4.40|4.45%|
> |HDMap prior module|66.12|66.87%|
> |Total |98.87| 100%|
> ## Weaknesses.4 & Questions.4:
> Indeed, we have also considered this aspect and supplemented our method with a cross-dataset experiment, detailed in **Appendix B.2** and **Table 11**. We initially pre-trained the HDMap prior module on the train split of the Argoverse 2 dataset and then tested it on nuScenes val dataset. This approach also yielded an improvement in the mIoU.

---

### Author Response · Authors · 2023-11-19
**Asking for feedbacks.**

Dear meta-reviewers and reviewers:

We express our gratitude to the meta-reviewers and reviewers for their valuable time dedicated to our manuscript. In response to these insightful feedback, we have diligently conducted new experiments and provided additional clarifications. We kindly request their consideration in reviewing these revisions at their earliest convenience. Your feedback on any remaining concerns would be immensely helpful, enabling us to offer further enhancements before the deadline on Nov. 22nd.

Sincerely, Authors.

---

### Meta-Review · Area_Chair_qwur · 2023-12-10

**Metareview:**

An online raster HMmapping method is proposed that incorporates an SDMap prior through a multi-head cross-attention module and HDMap priors with a ViT refinement module trained akin to MAE. The key strength of the paper is a well-presented and simple method that yields good results. They key weakness is an incomplete experimental evaluation.

All the reviewers agree that the paper is clear to understand and solves a well-motivated problem. The ablation experiment requested by Reviewer 8h86 are provided in the updated version. But the key issue remains the positioning with respect to recent works, where Reviewers Je8q and mZgt require a comparison to MapTR and MapTRv2, while Reviewer 6mPa also mentions comparison to VectorNet, PivotNet and MapVR. The author feedback includes a partial evaluation with respect to MapTR, but a more comprehensive set of comparisons is needed. Overall, while the author feedback and revisions to the paper add several ablation experiments, runtime analysis and robustness experiments, a more comprehensive comparison with respect to recent methods is needed and the overall extent of changes will require a new round of reviews. The AC agrees with the reviewer consensus that the paper may not be accepted for ICLR. The authors are encouraged to resubmit to a next venue with the complete set of experiments.

**Justification For Why Not Higher Score:**

Well-motivated paper, but significant gaps in experimentation which make the positioning with respect to recent works unclear.

**Justification For Why Not Lower Score:**

Not applicable.

---

### Decision · Program_Chairs · 2024-01-16

Reject